



# Line-averaging measurement methods to estimate the gap in the $CO_2$ balance closure – possibilities, challenges and uncertainties

Astrid Ziemann[1], Manuela Starke[1], Claudia Schütze[2]

[1]Chair of Meteorology, TU Dresden, Dresden, 01062, Germany
[2]Department for Monitoring and Exploration Technologies, Helmholtz Centre for Environmental Research, Leipzig, 04318, Germany

*Correspondence to*: Astrid Ziemann (astrid.ziemann@tu-dresden.de)

**Abstract.** An imbalance of surface energy fluxes using the eddy covariance (EC) method is observed in global measurement networks although all necessary corrections and conversions are applied to the raw data. Mainly during nighttime, advection can occur resulting in a closing gap that consequently should also affect the $CO_2$ balances. There is the crucial need for representative concentration and wind data to measure advective fluxes. Ground-based remote sensing techniques are an ideal tool as they provide the spatially representative $CO_2$ concentration together with wind components within the same voxel structure. For this purpose, the presented SQuAd (Spatially resolved Quantification of the Advection influence on the balance closure of greenhouse gases)-approach applies an integrated method combination of acoustic and optical remote sensing. The innovative combination of acoustic travel-time tomography (A-TOM) and open-path Fourier transform infrared spectroscopy (OP-FTIR) will enable an upscaling and enhancement of EC measurements. OP-FTIR instrumentation offers the significant advantage of real-time simultaneous measurements of line-averaged concentrations for $CO_2$ and other greenhouse gases (GHGs). A-TOM is a scalable method to remotely resolve 3D wind and temperature fields. The paper will give an overview about the proposed SQuAd-approach and first results of experimental tests at the FLUXNET site Grillenburg in Germany.

Preliminary results of the comprehensive experiments reveal a mean nighttime horizontal advection of $CO_2$ of about 10 µmol m$^{-2}$ s$^{-1}$ estimated by the spatially integrating and representative SQuAd method. Thereby, uncertainties in determining $CO_2$ concentrations using passive OP-FTIR and wind speed applying A-TOM are systematically quantified. The maximum uncertainty for $CO_2$ concentration was estimated due to environmental parameters, instrumental characteristics, and retrieval procedure with a total amount of approx. 30 % for a single measurement. Instantaneous wind components can be derived with a maximum uncertainty of 0.3 m s$^{-1}$ depending on sampling, signal analysis, and environmental influences on sound propagation. After the application of averaging over a period of 30 minutes, the standard error of the mean values can be decreased by a factor of at least 0.5 for OP-FTIR and 0.1 for A-TOM depending on the required spatial resolution. The presented validation of the joint application of the two independent, non-intrusive methods is in the focus of attention concerning their ability to quantify advective fluxes.



# 1 Introduction

A closing gap for energy balance measurements which affects the balance closure of greenhouse gases (GHGs), e.g. $CO_2$, is still observed at all stations in global measuring networks (Marcolla et al., 2014). This imbalance exists although all necessary corrections and calculations are applied to the measurement of flows using the eddy covariance (EC) method (e.g.

Foken et al., 2010, Mauder et al., 2006). Obviously, the existing measurement methods do not capture all relevant transport mechanisms, especially during calm and stable nighttime conditions. There has been a common agreement that EC measurements tend to underestimate carbon fluxes in such situations (e.g. Moncrieff et al., 1996, Baldocchi et al., 2000, Paw U et al., 2000). In this context, advection is an important mechanism. Advective fluxes can reach significant values, especially at low-turbulent conditions (Aubinet et al., 2003). That causes an uncertainty in the crucial determination of the

$CO_2$ mass balance of natural surfaces, e.g. forests. Thereby, an almost 50 % reduction of the estimated potential of forests as a carbon sink is possible (Siebicke et al., 2012). This uncertainty has an impact on the confidence level of climatological forecast models and consequently on the reliability of adaptation strategies to climate change (Richardson et al., 2012). Thus, the measurement of advection remains an important issue for accurate carbon sink or source estimates.

The following simplified equation for $CO_2$ mass conservation (NEE = net ecosystem exchange) includes the mentioned

advective fluxes and is commonly used e.g. within FLUXNET (e.g. Feigenwinter et al., 2008):

$$NEE = \int_0^{z_r} \frac{1}{V_m} \frac{\partial \overline{c(z)}}{\partial t} dz + \frac{1}{V_m} \overline{w'c'(z_r)} + \int_0^{z_r} \frac{1}{V_m} \frac{\overline{w(z)} \, \overline{\partial c(z)}}{\partial z} dz + \int_0^{z_r} \frac{1}{V_m} \left( \overline{u(z)} \frac{\partial \overline{c(z)}}{\partial x} + v(z) \frac{\partial \overline{c(z)}}{\partial y} \right) dz \tag{1}$$

with $V_m$ as the molar volume of dry air, $c$ is the $CO_2$ molar fraction (μmol mol$^{-1}$), $t$ is the time, $u$, $v$, and $w$ are the wind velocity components in $x$, $y$, and $z$ directions, respectively. Overbars indicate Reynolds averaging, typically over a time of 30 min.

The first term on the right-hand side describes the rate of change in storage of $CO_2$. The second term refers to the turbulent vertical flux which is usually measured as EC flux at the reference height $z_r$ above ground surface. Third and fourth terms are the non-turbulent vertical and horizontal advection terms, respectively. In practice, finite differences are used to approximate the spatial derivatives in Eq. 1. The horizontal advection at a reference height is simplified to:

$$F_{Hor} = \frac{1}{V_m} \left( \bar{u} \frac{\Delta \bar{c}}{\Delta x} + \bar{v} \frac{\Delta \bar{c}}{\Delta y} \right) \Delta z \tag{2}$$

where the wind components and horizontal concentration gradients are representative for a specific height layer $\Delta z$.

An equivalent equation could be derived for the vertical advection.

Experimental investigation of the advective $CO_2$ fluxes started in the late nineties (Lee, 1998). Several recent studies tried to quantify the effect of advection in the near surroundings of flux tower sites (e.g., Siebicke et al., 2012, Marcolla et al., 2014). The studies varied from 2D configurations (e.g., Aubinet et al., 2003) to more sophisticated 3D experimental designs (e.g.,

Feigenwinter et al., 2008).

Zeri et al. (2010) considered advective fluxes of $CO_2$ as high when they exceeded 5 μmol m$^{-2}$ s$^{-1}$. This value is in agreement with observations at other sites (Rebmann et al., 2004, Siebicke et al., 2012).





Advection occurs mainly in presence of flows associated with topographical slopes or with land use changes (Aubinet, 2008). Marcolla et al. (2014) measured within the experiment ADVEX during situations dominated by the local slope wind system positive horizontal and vertical advection (typical values around 7 and 3 µmol m$^{-2}$ s$^{-1}$, respectively) together with downslope winds at night and slightly negative horizontal advection (typical values around -2 µmol m$^{-2}$ s$^{-1}$) together with

upslope winds during the day. Taking such advective fluxes into account would significantly reduce the reported annual $CO_2$ uptake of the investigated forest.

A typical daily pattern of advection was described by several authors: advection is maximal after sunset, when higher gradients of $CO_2$ concentration are expected to occur with the onset of stable stratification (e.g. Kutsch et al., 2008). Siebicke et al. (2012) found an additional second maximum for stable stratification and low air temperature due to radiative cooling at

the end of the night. Sun et al. (2007) reported also significant horizontal $CO_2$ advection during transition periods in the early evening and early morning when turbulence intensity is low.

Advection measurements are mostly affected with large uncertainties (Rebmann et al., 2010). A big challenge is the accurate measurement of horizontal concentration gradients which are often small in relation to the measurement uncertainty (Heinesch et al., 2007). Additionally, a synchronous observation of horizontal gradients is not possible if several

measurement points are sequentially sampled. Because of the limited spatial resolution of observations, the spatial $CO_2$ concentration as well as the flow field is systematically under-sampled (Aubinet et al., 2010). This common limitation of point-based gradient measurements leads to an inadequate spatial and temporal sampling of the underlying phenomena (Marcolla et al., 2014).

Furthermore, advection is most likely a scale-overlapping process (Feigenwinter et al., 2010). The lack of knowledge of the

variability of scalar gradients in space and time has been identified as one of the most likely reasons inhibiting significant progress in solving the nighttime problem of underestimating carbon dioxide emissions from forested sites (Aubinet et al., 2010, Thomas, 2011). Marcolla et al. (2014) explained that the uncertainty due to the sampling in time and space with classical single point measurements can be two magnitudes larger at low measurement levels (i.e. at 0.5 m) in comparison to the instrumental uncertainty. The higher number of sample points in time and space results in a better temporal and spatial

averaging and reduces the impact of local effects (e.g., heterogeneous vegetation structure) on the 30-min averages derived by Siebicke et al. (2012). Horizontal and vertical resolutions of measurements as well as the size of the control volume are two crucial points for the experimental setup of actual sensor networks with multiple point measurements (Feigenwinter et al., 2010).

Another possibility to provide an adapted data sampling in space and time are line-integrating measurement methods which

are generally able to determine the required quantities of $CO_2$ advection. As one of the first examples, Leuning et al. (2008) used perforated tubing at several levels to perform line-integrated concentration measurements. But the combination of line-integrated concentration measurements with adequate and spatially representative measurements of wind components remained challenging (Siebicke et al., 2012). Consequently, the main objective of the current study is to develop and apply an adapted line-averaging method to measure wind components using acoustic tomography (A-TOM) and $CO_2$





concentrations applying open-path Fourier transform infrared spectroscopy (OP-FTIR). These methods are introduced in Sect. 2. The innovative combination of ground-based remote sensing methods was applied within the SQuAd project (Spatially resolved Quantification of the Advection influence on the balance closure of greenhouse gases) to quantify the distribution of $CO_2$ concentrations and wind vector in a consistent spatio-temporal resolution applying a special set-up and

analysis procedures (Sect. 3). A central point for further applications is the estimation of uncertainties of the proposed measurement and analysis methods including temporal and spatial resolution (Sect. 4). First results of nighttime measurements of horizontal advection over a grassland site are discussed and compared with typical values of other studies (Sect. 4, 5). Further developments and applications of the presented method combination are proposed in Sect. 5.

## 2 Line-averaging measurement and analysis methods

**2.1 Acoustic travel-time tomography A-TOM**

Acoustic travel-time tomography is a ground-based remote sensing technique that uses the dependence of sound speed in air on wind velocity and temperature along the sound path (Wilson and Thomson, 1994). Thereby, approximations are commonly applied to represent the sound speed in a moving medium considering an effective, motionless medium. The most common of these assumptions is the effective sound speed approximation (Rayleigh, 1945, Ostashev and Wilson, 2016):

$$\boldsymbol{c_{eff}}(T_{av}, \boldsymbol{v}) \; = \; c_L(T_{av}) + \boldsymbol{s} \cdot \boldsymbol{v_h} \quad \text{with} \quad c_L(T_{av}) = \sqrt{\gamma_d R_d T_{av}} \tag{3}$$

Here, $T_{av}$ is the acoustic virtual temperature. In addition to air temperature, $T_{av}$ accounts for effects of moisture on sound speed due to different molar masses of dry air and water vapour ($R_d = 287.05$ J/kg/K: specific gas constant of dry air) as well as their different ratios of specific heat capacities for constant pressure and constant volume ($\gamma_d = 1.4$: ratio of specific heat capacities for dry air). $\boldsymbol{v_h}$ is the horizontal wind velocity, $c_L$ is the sound speed for an adiabatic sound propagation after

Laplace (1816) depending on air temperature and air moisture, and $\boldsymbol{s}$ is the unit vector tangential to the sound ray path between sound source and receiver. For sound propagation near the ground with small elevation angles (Ostashev and Wilson, 2016), the effective sound speed is often used in the form:

$$c_{eff}(T_{av}, v_h) \; = \; c_L(T_{av}) + v_h \cdot cos\ \varphi = c_L(T_{av}) + v_{Ray}, \tag{4}$$

where $\varphi$ is the angle between the azimuthal direction of sound propagation and the horizontal wind speed $v_h$, and $v_{Ray}$ is the

wind speed in direction of sound propagation.

Effective sound speed can be estimated from travel-time measurements:

$$\tau = \int_{x_S}^{x_R} \frac{ds}{c_{eff}} \qquad , \tag{5}$$

where $\tau$ is the acoustic travel-time of a signal propagating along a sound path with distance elements $ds$ between sound source at position $x_S$ and receiver at $x_R$. Travel-time measurements of acoustic signals propagating along different paths

through an air volume give information on the spatial distribution of sound speed within the investigated area. Exactly



knowing positions of loudspeakers and microphones, spatial distributions of flow and temperature fields can be reconstructed applying inverse algorithms (e.g. Ostashev et al., 2009). As a remote sensing method, one advantage of acoustic travel-time tomography is its ability to measure the meteorological quantities without disturbing the area under investigation due to insertion of sensors. The scalable method enables inertia-free measurements without influences of radiation on the sensor. Furthermore, temperature and wind speed can be recorded simultaneously with this measurement method (Vecherin et al., 2006).

Acoustic tomography as a measurement and analysis method has been further developed since the late 1990s (Ziemann et al., 1999, Arnold et al., 2001). This method was used to monitor spatially resolved wind and temperature fields for different environmental conditions, e.g. in rural (Ziemann et al., 2002) or urban environment (Tetzlaff et al., 2002) with heterogeneous surface properties (Raabe et al., 2005) as well as on different spatial scales, from indoor wind tunnel length scales (Barth and Raabe, 2011, Barth et al., 2007) up to outdoor areas with acoustic path lengths of several 100 m (Arnold et al., 2004). Thereby, several inversion techniques were developed and validated regarding their potential for special applications (Fischer et al., 2012). First joint investigations using A-TOM and optical spectrometers confirmed the suitability of combined line-integrating measurements of GHG exchange between surface and atmosphere (Barth et al., 2013, Schäfer et al., 2012).

The performance of A-TOM to reconstruct wind and temperature fields depends on several factors (Ziemann et al., 2007):

(1) the accuracy of travel-time estimates which is influenced by the signal characteristics (e.g. frequency, kind of signal) and the method of data analysis (correlation technique) and (2) the sound path length and its uncertainty due to sound propagation effects, especially refraction and reflection of sound waves, as well as positioning accuracy of sound sources and receivers.

Thus, the setup of the A-TOM measurements (e.g. positioning of loudspeakers and microphones to optimize the signal-to-noise ratio, SNR) determines the accuracy of the wind components for the calculation of advection. A detailed treatment of uncertainties is given in chapter 4.1.

## 2.2 Open path Fourier transform infrared (OP-FTIR) spectroscopy

The open, unobstructed atmosphere can be described as a complex, multi-component system controlled by parameters such as wind, temperature variation, rain, and pressure fluctuations. The driving parameters for the infrared (IR) transmittance of the atmosphere are the presence and the concentration of gas molecules and the length of the optical pathways. The interactions between IR energy and molecules cause characteristic absorption or emission lines in the measured spectra (Griffiths and de Haseth, 2007). The concentration of gases along the optical pathway can be retrieved by using the Beer-Lambert law. Open-path technology concepts are applied to measure the absorption loss along an optical path in ambient air. For passive measurements, changes in the main infrared atmospheric window with respect to absorbing gases are recorded. For active systems, an IR beam is transmitted through open, unobstructed atmosphere and the measurement obtained represents an integrated gas concentration along the optical path - so called 'path integrated concentration values – PIC'



(DIN EN 15483, 2009). The transmissivity of the atmosphere is more or less controlled by the presence of $H_2O$ und $CO_2$. Hence, due to the strong interference with water and carbon dioxide we can identify three main spectral windows available for OP-FTIR measurements in wavenumber regions ($\nu$): (1) 700 – 1300 $cm^{-1}$ (passive / active OP-FTIR), (2) 1900 – 2250 $cm^{-1}$, and (3) 2400 – 3000 $cm^{-1}$ (window 2 and 3 only for active OP-FTIR usable; Marshall et al., 1994).

The wavenumber-dependent IR intensity after passing through an absorbing sample $I(\nu)$ can be described by

$$I(\nu) = I_0(\nu) \cdot \exp\{-t(\nu)\} \qquad (6)$$

where $I_0(\nu)$ is the IR intensity emitted from IR source and $t(\nu)$ represents the optical depth, a sum function over all absorption lines $\alpha_i$ multiplied with concentration $c_i$ (substance amount) of the molecules $i$ and the path length $d$:

$$t(\nu) = d \cdot \sum_i \alpha_i(\nu)c_i = d \cdot \alpha_T(\nu) \cdot c_T + d \cdot \sum_{i-1} \alpha_{i-1}(\nu)c_{i-1} . \qquad (7)$$

The optical depth $t(\nu)$ includes the absorption behavior of the target molecule ($\alpha_T$, $c_T$) as well as the influence of all interfering atmospheric molecules along the measured optical path $d$.

Hence, Eq. (6) can be written as

$$I(\nu) = I_0(\nu) \cdot exp\left\{-d \cdot \sum_{i-1} \alpha_{i-1}(\nu)c_{i-1}\right\} \cdot exp\{-d \cdot \alpha_T(\nu) \cdot c_T\}$$

$$= I_0^*(\nu) \cdot exp\{-d \cdot \alpha_T(\nu) \cdot c_T\}. \qquad (8)$$

The expression $I_0^*(\nu)$ represents the background spectrum including absorption due to all disturbing molecules.

OP-FTIR spectroscopy is proven to be a powerful technique enabling online monitoring of fugitive emissions for industrial, environmental and health applications (e.g., Harig and Matz, 2001, Griffith et al., 2002, DIN EN 15483, 2009). It allows spatial characterization of emissions and can be applied non-invasively as an automated surveillance method in large and potentially inaccessible areas (Schütze et al., 2015). Furthermore, ground-based optical remote sensing methods like OP-FTIR are well suited to study dynamic atmospheric processes due to their avoidance of any disturbances upon emission

and/or sampling processes (Reiche et al., 2014, Schütze and Sauer, 2016). Several successful applications of active and passive OP-FTIR are reported in terms of air quality monitoring, dynamic atmospheric processes observations and emission rate estimations in boundary layer (e.g., Griffith et al., 2002, Allard et al., 2005, Schäfer et al., 2012, Chen et al., 2015). Then, the technique is often combined with other micrometeorological investigations and provides information on several GHG target gases, such as $CO_2$, $CH_4$, $NH_3$, and $N_2O$ (Griffith et al., 2012, Wilson and Flesch, 2016). Flesch et al. (2016)

emphasize the potential of combined micrometeorological and OP-FTIR measurements for enhanced GHG emission determinations.

The determined gas concentrations base on the retrieval of concentration values from measured IR spectra. The concentration value obtained is associated with an uncertainty that characterizes the dispersion due to random and systematic errors caused by the measurement and the data processing procedures (Schütze and Sauer, 2016). Thus, instrumental

characteristics, applied infrared sources, environmental parameters, and retrieval algorithms represent the main sources of uncertainty. The assessment of uncertainties for these influencing factors relating to the Grillenburg experiment will be discussed in chapter 4.2.




## 3 Site, experimental set-up and data analysis

### 3.1 Grassland EC site Grillenburg

A joint experiment with A-TOM and OP-FTIR techniques as well as additional measurement equipment was carried out within the SQuAd project at the EC site Grillenburg. The grassland test site (380 m a.s.l., 50 °57 ′04 ″ N, 13 °30 ′50 ″ E) is located in the middle of a large clearing (40 hectare area) within the Tharandt Forest, 30 km away from Dresden in Germany (Figure 1).

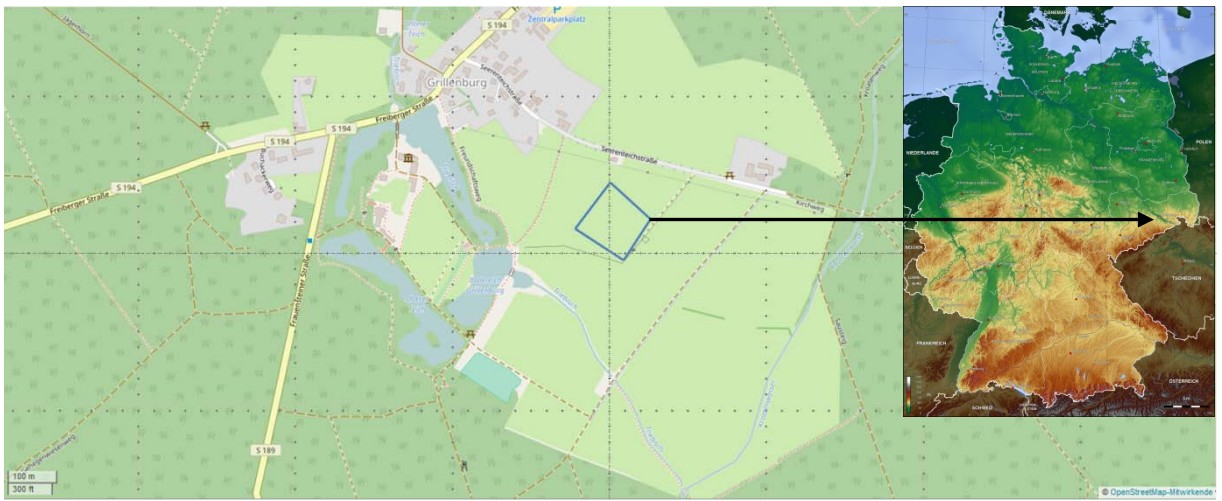

**Figure 1: Map of Grillenburg, meadow in light green, Tharandt Forest in dark green, area under investigation is marked by the blue rectangle (Source of maps: Grillenburg area: OpenStreetMap, https://www.openstreetmap.org/#map=16/50.9501/13.5122, Germany: http://www.mygeo.info/landkarten/deutschland/Deutschland_Topographie_2007.jpg, GNU free documentation license)**

An eddy flux tower was established there at a meadow which is extensively managed with 2 to 4 hay harvests per year. The mesophytic hay meadow is dominated by couch grass (Agropyronrepens), meadow foxtail (Aleopecurus pratensis), yarrow (Achillea millefolium), common sorrel (Rumex acetosa), and white clover (Trifolium repens) (Prescher et al., 2010). Cows, sheep or horses were rarely grazing there. Neither mineral nor organic fertilizers have been applied at this site since 1987. The permanent EC station is working within FLUXNET since 2002 (e.g. Hussain et al., 2011a). Meanwhile it is a part of ICOS-D (C3 station). Grillenburg is an atmospheric carbon sink (Prescher et al., 2010). However, the NEE values show large inter-annual differences (e.g. −177 g C m$^{-2}$, 2006 to −62 g C m$^{-2}$, 2005). The mean Net Ecosystem Productivity (NEP) is about 80 gC m$^{-2}$ a$^{-1}$ since 2005. After incorporation of carbon export due to harvest of hay the permanent grassland becomes a $CO_2$ source of about 60 gC m$^{-2}$ a$^{-1}$ (ICOS-D Website, 2017).

The station is actually equipped with the following EC measurement technique to determine turbulent $CO_2$- and $H_2O$-fluxes at a height of 3 m above ground: ultrasonic anemometer GILL R3-50 (Gill Instruments, Lymington, UK) and close-path measurements with IR gas analyzer (IRGA) LI-7000 (LiCor Inc., Lincoln, NE, USA). Fluxes of $CO_2$ (NEE), $H_2O$





(evapotranspiration) and sensible heat are available on a half-hourly basis. Based on the general EUROFLUX guidelines (Aubinet et al., 2000), Grünwald and Bernhofer (2007) described the calculation and correction of the fluxes which are permanently updated according to sensor and software development.

Additionally, air temperature and air humidity, soil temperature and soil heat flux, global and net radiation, photosynthetically active radiation as well as precipitation and evaporation (class-a-pan) are measured at the station permanently.

The nearby climate station is delivering data since 1862 and is at the same location since 1955. The annual mean temperature is 7.8 °C and the annual mean precipitation is 901 mm (period 1981-2010).

## 3.2 Experimental set-up in July 2016

The special observation period (SOP) was carried out shortly after the harvest of hay in Grillenburg from 8[th] until 18[th] of July in 2016. Two periods were of special interest because of high solar radiation during the day (convective boundary layer) and the building up of a stably stratified boundary layer during nighttime: 9[th] until 11[th] and 15[th] until 18[th] of July.

On the 8[th] of July, shortly after the set-up, the test site was affected by a thunderstorm. Therefore, the measurements started on the next day. At this time the area of investigation was influenced by a high ridge whose axis was directed from north to south across the center of Germany on the 11[th]. In its northern part the ridge was overrun by strong warm air advection due to an upper air trough which travelled eastward towards Ireland. Within the broad-based warm sector, very warm air masses from the southwest influenced the experimental site especially on the 10[th] and 11[th]. The air mass was potentially unstably layered but was also strongly capped due to the low-tropospheric warm air advection. After this, the weather conditions changed to rainy days due to a trough over central Europe which led to a break of the measurements. On 15[th] of July a high ridge from the Biscay to the North Sea started to influence the weather conditions. Initially, fairly moist and cool air reached the area with a north-westerly wind direction. The following days were characterized by an intermediate high.

In order to obtain statements on advection during the SOP, information on spatially distributed $CO_2$ concentrations was estimated from scanning passive OP-FTIR devices. For continuous calibration, two active OP-FTIR devices have been applied. A-TOM was configured in such a way that spatially averaged wind velocities could be measured in two different heights above the ground (1.5 m, 3.0 m). As reference for the line-averaged A-TOM measurements, two masts, each of which equipped with two ultrasonic anemometers (Young) at two heights (1.5 m and 3.0 m), were arranged at the side of the A-TOM measurement area (see Fig. 2).

The total area under investigation, approx. 120 m x 120 m, is marked in Fig. 1. All locations were measured using GPS and by a high-precision theodolite.

The acoustic measuring field is limited by the position of the acoustic devices, which are mounted on telescopic masts at the corners of the field (ATOM1…4). The height difference within the acoustic measuring field (Fig. 2) is about 2.2 m, estimated from own tachymeter measurements. The terrain rises in northern direction from the EC station (near to ATOM1)



to the location of mast ATOM3. Between the masts equipped with ultrasonic anemometers (Young1 and Young2, horizontal distance of 65 m), the difference in terrain height is approx. 0.5 m.

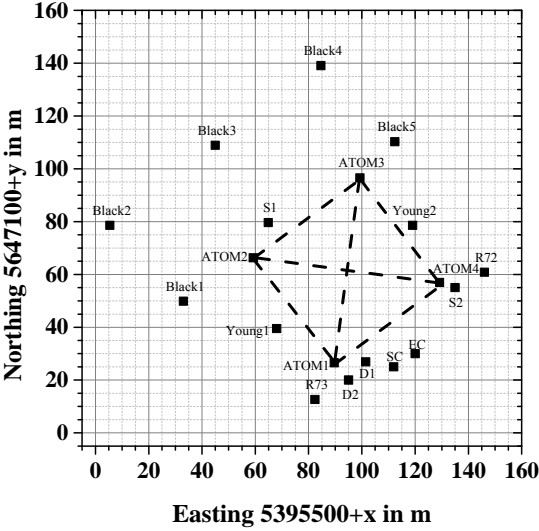

**Figure 2: Scheme of area under investigation (see also Fig. 1) with location of several devices and auxiliary equipment: ATOM1-4 (4 masts for travel-time tomography A-TOM), dashed lines mark acoustic paths, R72/73 (2 Bruker Rapid, passive OP-FTIR), D1/2 and S1/2 (2 Bruker EM27, active OP-FTIR with source and detector), Young1/2 (2 masts, each equipped with 2 ultrasonic anemometers), Black1-5 (5 black screens for passive OP-FTIR), EC tower, SC soil respiration chamber measurements.**

### 3.2.1 Set-up wind measurements

The A-TOM area inside this field extended to about 50 m x 50 m (Fig. 2). For wind velocity estimation, a tested tomographic measurement system was adapted to the proposed measurements at two height levels (1.5 m and 3 m). Four high-end horn speakers for frequencies above 5 kHz (TL16H, 8 Ohm, Visaton) and four free-field prepolarized microphone units (½", Type 4189-A-021, Brüel&Kjær) with windscreens were built up at telescopic masts with special booms at both heights above ground surface. Thus, each mast was equipped with two loudspeakers and two microphones (Fig. 3).

For typical sound speeds of 340 m s$^{-1}$ the maximum travel-time for the introduced A-TOM set-up was 0.2 s due to the maximum length of sound paths of about 70 m. The time interval between successive measurements (estimation of travel-time data along all relevant sound paths) was 20 s due to the duration of signal analysis and data storage of all 24 single measurements (12 at each height level: i.e. forth and back between ATOM1-2, 1-4, 1-3, 2-3, 2-4, 3-4, see Fig. 2).





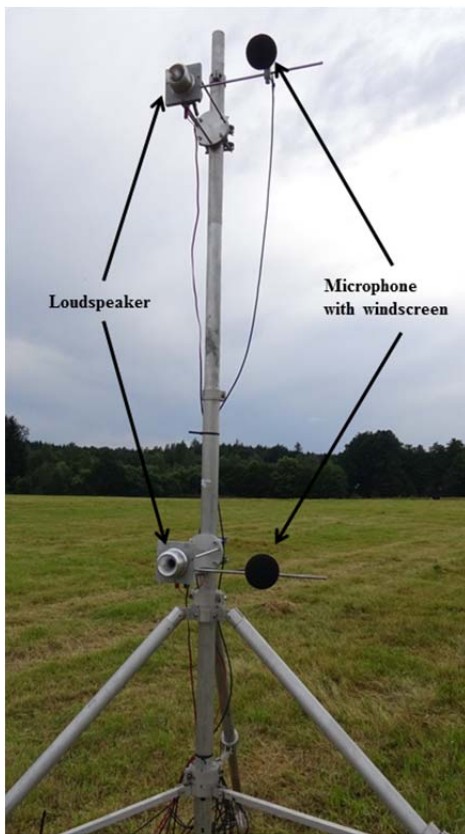
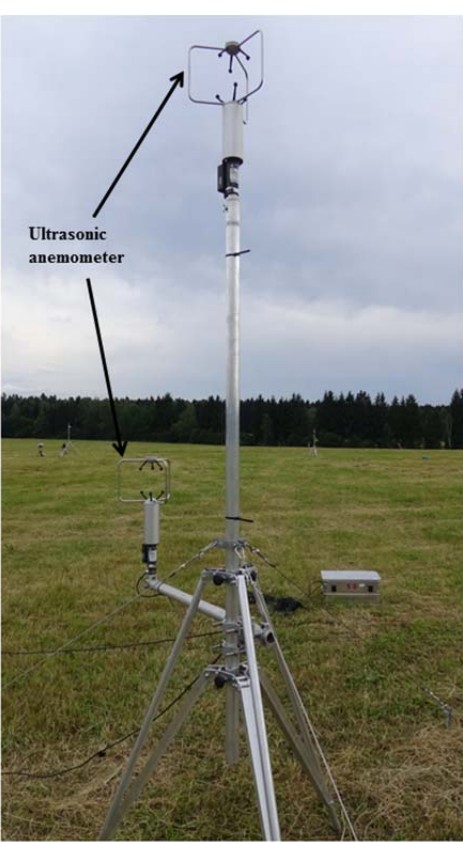

**Figure 3: Left: Telescopic mast (A-TOM2) with acoustic equipment at two height levels in 1.5 m and 3 m above grassland, single tree line 220 m away and Tharandt Forest at a distance of 450 m in the background (southwest direction). Right: Telescopic mast with ultrasonic anemometers at 1.5 m and 3 m height (Young2).**

The described acoustic system can be enhanced in future experiments with additional sound sources and receivers to increase the spatial resolution of the measurements which is especially desirable for the application of tomographic data analysis.

The four supplementary ultrasonic anemometers (YOUNG8100V, R. M. Young Company, Michigan, USA) were mounted side by side at a height of 2.26 m above ground at the EC station Grillenburg for a period of 6 days ($10^{th}$ – $16^{th}$ June) shortly

10    before the SOP. The obtained data were compared among each other to guarantee that all devices are measuring the same value, a requirement to calculate vertical or horizontal gradients with high accuracy. Although all anemometers are of the same kind, series and age, there are differences in acoustic virtual temperature due to the special characteristics of the individual instrument. One anemometer was used as reference. Regressions between the temperature data of the reference and the other devices were calculated. These equations were used during the SOP to correct the measured temperature values

15    of the ultrasonic anemometers. For the wind velocity, the quantity of primary interest, such a correction was not necessary.





### 3.2.2 Set-up concentration measurements

Successful application of the non-intrusive methods A-TOM and OP-FTIR requires agreement in the investigated air volume and the spatial resolution of trace gas concentration and wind components. Thus, the OP-FTIR technique was built up within and around the A-TOM array (Fig. 2).

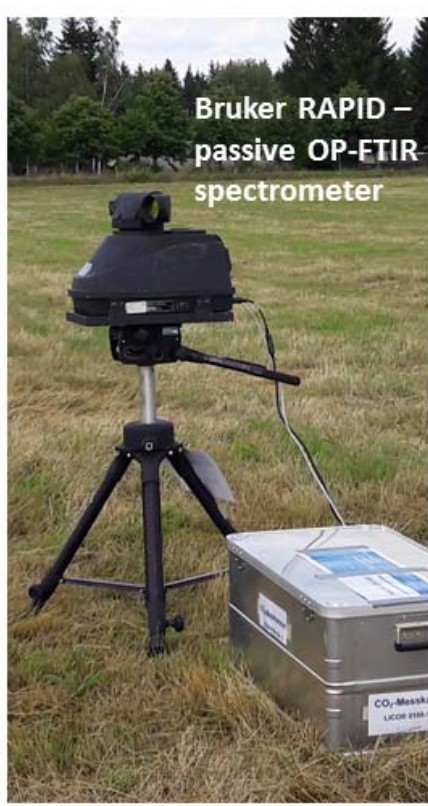
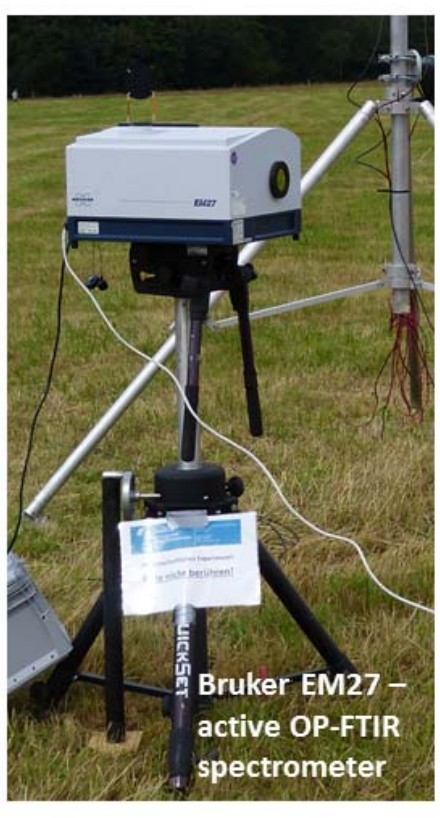

**Figure 4: OP-FTIR spectrometer used for SOP at Grillenburg site. Left: passive Bruker RAPID, right: active Bruker EM27 detector unit applied in bi-static mode with a separate IR source (not in figure).**

For our OP-FTIR investigations (Fig. 4) we used two BRUKER EM 27 systems (Bruker Optik GmbH, Ettlingen, Germany)
10  in bistatic operation mode including NiCr glowers as field IR source for active measurements and two BRUKER RAPID spectrometers (Bruker Daltonik GmbH, Leipzig, Germany) for passive investigations. Both devices include narrow-band MCT (mercury cadmium telluride) detectors and can be characterized by its instrumental parameters (Table 1).

A detailed description of equipment characteristics for both devices is listed by Schütze and Sauer (2016).

The installation of the spectrometers and associated instruments (sources, screens) was undertaken avoiding any influences
15  on micrometeorological and acoustic measurements. Furthermore, the optical pathways had to be aligned without obstructions. The active OP-FTIR measurements were carried out on two perpendicular aligned optical paths situated in



close vicinity to the A-TOM equipment (Fig. 2). The two EM27 spectrometers and their associated IR sources were installed obtaining optical path lengths of 52 m and 64 m, respectively. The spectral measurements were carried out in 2 minutes sampling intervals including a co-addition of 20 spectra to improve the signal-to-noise ratio (SNR).

**Table 1: OP-FTIR spectrometer device parameters.**

| Instrumental parameter | Bruker RAPID | Bruker EM27 |
|---|---|---|
| modus | Passive | Active / passive |
| IR source | Ambient | NiCr glower at 1200 °C |
| detector | MCT | MCT |
| resolution | 4 cm$^{-1}$ | 1 cm$^{-1}$ |
| Field of view (FOV) | 10 mrad | 10 mrad |

For passive measurements the two RAPIDs were installed at the outer edges of the field of investigation in a distance of 80 m from each other. Five black background screens were used as potential targets for the passive measurements. A complete measurement consisted of 12 different single beam acquisitions with 6 different horizontal directions per device

aiming at an even distribution of optical path ways inside the field of investigation. The sampling interval was 5.5 minutes. For each measurement an internal temperature controlled black body within the spectrometer device was applied as a defined radiation source to calibrate the instrument.

In order to obtain information on ground surface $CO_2$ concentration and soil emission, a LI-COR 8100 system including a multiplexer LI-8150 and two long-term chambers were installed nearby the EC tower (Fig. 2). We chose a sampling interval

of two measurements per chamber per hour for the data acquisition period. The chambers installation was done one day before acquisition started to avoid any influences by disturbances due to the collar insertion. The obtained $CO_2$ data can be applied for the comparison with the spatially resolved GHG concentrations.

### 3.3 Data analysis

### 3.3.1 Signal processing and analysis of acoustic travel-time measurements

The acoustic measurements are controlled by an in-house developed software (MATLAB) which comprises generation of sound signals, control of sound transmission and reception, as well as subsequent real-time signal analysis. The core hardware (analog/digital conversion) is an acoustic multi-channel spectrometer card (Harmonie PCI octav, sample rate: 51.2 kHz, SINUS Messtechnik GmbH, Germany) which offers 8 input and 4 output channels that are synchronized on a common time basis (Holstein et al., 2004, Barth and Raabe 2011). This, in turn, is a precondition for accurate travel-time

measurements.





Acoustic signals with a frequency of 7 kHz and a special signature (sine signal with 2x2 oscillations and a break between them, Fig. 5) are used. The 7-kHz-signal is a compromise between the desired low travel-time uncertainty and the necessary high SNR. First requirement is fulfilled with growing sound frequencies. The last one requires minimal environmental effects, especially sound absorption in air which increases with growing frequency. Furthermore, higher frequencies allow for a high-pass filtering of received signals in order to exclude ambient low-frequency noise from data analysis which, in turn, enhances SNR. Considering in addition acoustic ground surface effects (see Sect. 4.1.2), an optimal sound frequency of 7 kHz results for the investigated length scale up to 100 m.

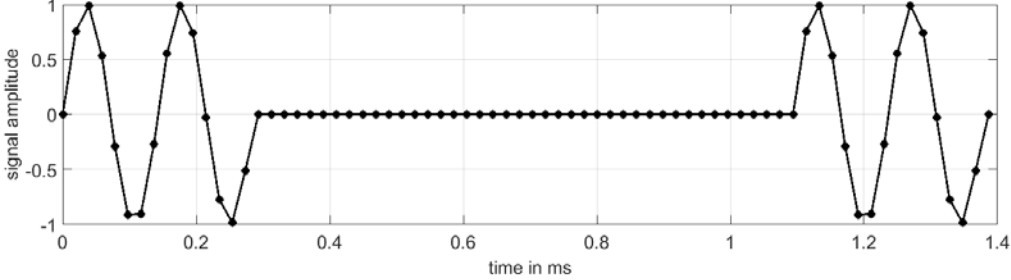

**Figure 5: Theoretical acoustic signal consisting of two times two sine periods with a frequency of 7 kHz interrupted by a break. The sample rate of the analog-to-digital converter is 51.2 kHz.**

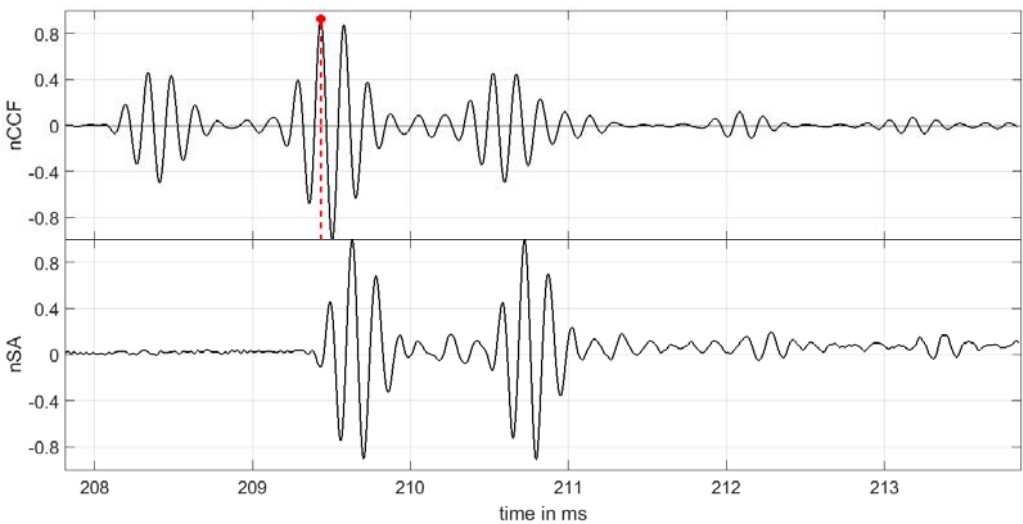

**Figure 6: Example of a received acoustic signal (normalized signal amplitude, nSA, lower panel, distance of source and receiver: 70.45 m) and corresponding normalized cross-correlation function (nCCF, upper panel) between the received and the generated signal. The maximum position of nccf is marked with a filled (red) point. The associated time lag corresponds to the travel-time of the signal.**

After propagating through the atmosphere, the sound signal was received by the microphones, and high-pass filtered. The analog acoustic signals are sampled by the acoustic spectrometer card with a sample rate of 51.2 kHz, i.e. with a time





resolution of 19.5 µs. The delay time between output and input channels is known and constant and can be therefore neglected in the further analysis of accuracy. Subsequently, the sent signal is cross-correlated with the received signal. The maximum of the cross-correlation function (CCF) corresponds to the best fit of the sent signal pattern within the received signal. The associated time-shift corresponds to the sought travel time (Hussain et al., 2011b; Fig. 6).

To increase the accuracy of the detected maximum, an interpolation with a sinc function is applied which leads to an increased temporal resolution by a factor of about 10. Thus, an uncertainty for travel-time estimation of about 2 µs results from sampling (Holstein et al., 2004).

The A-TOM masts mark the corners of a rectangle at each level above surface (see Fig. 2). In order to separate the scalar influence of temperature and the vectorial influence of wind velocity on the speed of sound between a source and a receiver

(Eq. (4)), sound propagation is considered in opposing directions. Similar to the analysis of ultrasonic measurements (e.g. Hanafusa et al., 1982), the assumption of reciprocal sound propagation (straight ray propagation between two pairs of speakers and microphones) is applied:

$$c_{eff_{forth}} = \frac{d}{\tau_{forth}} = \sqrt{\gamma_d R_d T_{av}} + v_{ray} \,, \quad c_{eff_{back}} = \frac{d}{\tau_{back}} = \sqrt{\gamma_d R_d T_{av}} - v_{Ray} \,. \tag{9}$$

Here, $d$ is the distance between sound source and receiver, $v_{Ray}$ is the wind component in direction of sound propagation (cp.

Eq. 4), and $\tau_{forth}$ and $\tau_{back}$ are estimated travel times in opposing directions. If the distance $d$ is known, it follows from Eq. (9):

$$\sqrt{T_{av}} = \frac{d}{2\sqrt{\gamma_d R_d}} \left( \frac{1}{\tau_{forth}} + \frac{1}{\tau_{back}} \right) \quad \text{and} \tag{10}$$

$$v_{Ray} = \frac{d}{2} \left( \frac{1}{\tau_{forth}} - \frac{1}{\tau_{back}} \right). \tag{11}$$

In this way, the derivation of wind components along six sound paths as line-averaged data set is possible. Wind components

$u$ (in east direction) and $v$ (in north direction) are calculated from Eq. (11) for each of two sound paths approximately perpendicular to each other (e.g. path between ATOM1-ATOM2 and ATOM1-ATOM4, Fig. 2).

### 3.3.2 Spectral data acquisition and processing of OP-FTIR measurements

The passive and active IR spectrometer systems were linked with their own controlling laptops using OPUS software (Bruker Optics Inc.). The software provides interfaces to control measurement options such as spectral region for

measurement, wavenumber resolution, parameters for discrete Fourier transform, apodization function, and repeat intervals. Additionally, for passive measurements a user-written macro-program is necessary for controlling the instrument. This macro contains the detailed measurement sequence for a whole passive scan including the parameters for the preceding internal blackbody measurements and the acquisition parameters for the different scans (number of scan directions, vertical and horizontal lens angle, repetition rate).





An OP-FTIR spectroscopic measurement results in a single beam spectrum (SBS). It describes the distribution of signal intensity with respect to the wavenumber. The active SBS covers a wavenumber region between 600 – 3900 cm$^{-1}$, the passive SBS ranges between 600 – 1600 cm$^{-1}$. Subsequent data processing of SBSs is necessary for concentration analysis. In practice, the spectra obtained by spectrometer device are controlled by instrumental line shape $f_{ILS}$

$$I'(v) = I(v) \otimes f_{ILS}(v) \tag{12}$$

where $\otimes$ represents convolution.

A transmission spectrum $TR(v)$ of the sample can be obtained by dividing the measured spectrum $I'(v)$ by the measured or simulated background spectrum $I_0^{*'}(v)$ which is also influenced by $f_{ILS}$:

$$TR(v) = \frac{I'(v)}{I_0^{*'}(v)} \quad . \tag{13}$$

The absorbance spectrum $A(v)$ of the target component is introduced as a linear function related to target compound concentration:

$$A(v) = -log_{10}\big(TR(v)\big) = 0.4343\, d \cdot \alpha_T(v) \cdot c_T \quad . \tag{14}$$

The crucial difference between active and passive OP-FTIR measurements results from the availability of different $I_0(v)$ sources:

- active: superposition of non-modulated artificial IR source (wavenumber region 700 – 4000 cm$^{-1}$) and additional ambient (passive) background emissions for wavenumbers lower than 1500 cm$^{-1}$
- passive: only ambient background emissions resulting from black body radiation according to Planck's law limited to wavenumber region between 700 – 1500 cm$^{-1}$. This emission is a function of radiometric temperature (temperature of the IR emitting surface).

The data processing of active spectra includes the emission correction of SBSs for lower wavenumber regions, the calculation of transmission spectra based on reference spectra, and the determination of spectral windows for $CO_2$ concentration analysis. The concentration retrieval uses a non-linear least square fitting of measured by calculated spectra using HITRAN spectral library (Rothman et al., 2013).

The processing of passive spectral data is different compared to active spectra. Passive OP-FTIR measures radiation from background traversing the atmosphere between the background and the spectrometer. The black body radiation $B(v,T)$ can be described according to Planck's radiation law

$$B(v,T) = \frac{2\,h\,c^2 v^3}{exp\left(\frac{h\,c\,v}{k_B T}\right)-1} \tag{15}$$

where $h$ is Planck constant, $c$ is speed of light, $k_B$ is Boltzmann constant. In order to obtain radiance spectra or brightness temperature spectra a radiometric calibration of SBSs is necessary. This calibration algorithm is based on the SBS measurement of an ideal black body within the spectrometer device at two known temperatures: $T_C$ = ambient temperature and $T_H$ = 353 K. The radiance spectra $L(v,T)$ of a measured $SBS(v,T)$ can be obtained after Revercomb et al. (1988) from

$$L(v,T) = \frac{SBS(v,T)-SBS(v,T_C)}{SBS(v,T_H)-SBS(v,T_C)}\big(B(v,T_H)-B(v,T_C)\big) + B(v,T_C). \tag{16}$$


The determination of transmission spectra $TR(v)$ requires a radiative transfer model that includes the radiance of the background $L_B(v, T_B)$ as well as the self-radiance of the considered air volume $B(v, T_{air})$ (Liu et al., 2008):

$$TR(v) = \frac{L(v,T) - B(v, T_{air})}{L_B(v, T_B) - B(v, T_{air})} = \frac{L(v)}{L_0(v)} \, . \tag{17}$$

Similar to the active spectra processing, the calculated transmission spectra can be analyzed to obtain PIC values based on

the minimization of the difference between measured and simulated spectra. For both OP-FTIR techniques the non-linear relation between spectral signature of the target gas and its column density is used for the quantification. The radiative transport model and the influences of the applied spectrometer are required input parameters. The column density is the unknown model parameter. The forward modeling approach is based on the calculation of synthetic spectral windows (10 – 100 cm$^{-1}$) including the consideration of multiple parameters such as column density for each species (including additional

atmospheric substances), background radiation, temperature, pressure, and instrumental line shape functions. In the next step the synthetic and measured spectral windows are compared. Least-squares fitting algorithms (e.g., classical least squares regression CLS, partial least squares regression PLS) are applied in order to minimize iteratively the difference between both of them (Harig and Matz, 2001, Griffith et al., 2012, Cieszczyk, 2014).

## 4 Results and discussion

### 4.1 Uncertainty of A-TOM-wind and temperature measurements

### 4.1.1 Accuracy of travel-time estimates

Technical, signal-dependent, and methodological issues influence the travel-time determination leading to uncertainties due to sampling, signal analysis and cross-correlation, calculation of sound speed, and recalculation of wind speed and temperature.

First, the SNR should be as high as possible. Thus, disturbing sound near the microphones should be avoided. The flow field itself leads to the most important disturbance. With the used windscreens a maximum wind speed of about 6 m s$^{-1}$ is desirable without a noticeably changed characteristic of microphone sensitivity. Otherwise, higher efforts are necessary to protect the microphones against environmental sound.

It was explained in Sect. 3.3.1 that the analog signal is sampled with a sample rate of 51.2 kHz (time resolution of 19.5 μs).

The travel-time estimation is improved by using an interpolation technique which results in an uncertainty of about 2 μs for the travel-time data from sampling algorithm.

The period duration of a 7-kHz-signal is 1/7000 Hz≈143 μs, i.e. about 51.2 kHz/7 kHz≈7.3 samples for a digitization frequency of 51.2 kHz. Neighbouring maxima of the CCF are separated by about 7 samples. To rate this value it is helpful to calculate the typical travel-time variations (i.e. $\Delta \tau$) in sample units due to variability of meteorological data (Table 2). A

change in temperature of 1 K results (for a windless atmosphere) in a variation of about 0.6 m s$^{-1}$ in sound speed (see Eq. (3)). In comparison to that, the variations in wind speed (wind component along sound path) result in equal changes of sound





speed. If there are variations in both quantities, temperature and wind speed, the effects on the effective sound speed are summed up according the Eq. (4).

**Table 2: Variability of acoustic travel time (in rounded sample units) due to changes of temperature and wind speed for a mean temperature of about 8 °C.**

| Distance source-receiver in m | Temperature variation of | | Wind speed variation of | |
|---|---|---|---|---|
| | 0.5 K | 1.0 K | 0.5 m s$^{-1}$ | 1.0 m s$^{-1}$ |
| 50 | 7 | 13 | 11 | 22 |
| 70 | 9 | 18 | 15 | 30 |

To decrease the uncertainty due to analysis of CCF, it is possible to use the maximum of CCF's absolute value. In this way, the neighbouring maxima are separated only by about 3.7 samples. This value for the travel-time accuracy of 78.125 μs (= 4 samples/51.2 kHz) is applied for the further uncertainty analysis of sound speed, wind speed, and temperature for one instantaneous travel-time measurement along one sound path.

The influence of a faulty variable $x_i$ on the result $y$ can be estimated by means of the Taylor series. If the absolute value of error $\Delta x_i$ is small enough, the Taylor series can be aborted after the linear term resulting in an estimation of maximum error $\Delta y = \sum_i \left| \frac{\partial y}{\partial x_i} \right| \Delta x_i$. The complete derivation of temperature and wind uncertainty is shown in Appendix A. It results from Eq. (29):

$$\Delta T_{av} = 2 \sqrt{\frac{T_{av}}{\gamma_{tr} R_{tr}}} \Delta \tau \left( \frac{(\gamma_{tr} R_{tr} T_{av}) + v_{Ray}^2}{d} \right) \qquad . \tag{18}$$

For a travel-time accuracy of 78.125 μs and a minimal path length of 50 m a maximum temperature uncertainty of about 0.3 K results for the instantaneous single path measurement. The uncertainty of relative wind measurements is only depending on the uncertainty of travel-time measurements. Assuming again that travel-time errors along one and the same path are identical, it follows from Eq. (30):

$$\Delta v_{Ray} = \Delta \tau \left( \frac{(\gamma_{tr} R_{tr} T_{av}) + v_{Ray}^2}{d} \right) \qquad . \tag{19}$$

For a minimal path length of 50 m results a maximum wind component uncertainty for the instantaneous single path measurement of about 0.2 m s$^{-1}$. With increasing path lengths, the uncertainty of temperature and wind components is decreasing.

Considering these uncertainties as standard deviation of a single measurement, the standard error of mean values decreases by the factor $1/\sqrt{n}$ if the measurement is repeated $n$-times under one and the same boundary and environmental conditions.





Applying averaging over 30 minutes (90 independent measurements, i.e. $1/\sqrt{90} = 0.1$) results in statistical uncertainties of 0.03 K and 0.02 m s$^{-1}$, if all single measurement results are usable.

### 4.1.2 Accuracy of sound path estimation

The uncertainty of line-averaged wind and temperature data is further influenced by several effects of the sound propagation

between a loudspeaker and a microphone: absorption in air, reflection at ground surface, and refraction due to wind and temperature gradients.

A point source generates spherical waves in an unbounded homogeneous atmosphere (e.g. Salomons, 2001). In this simple case the sound pressure level at a microphone can be calculated from the sound power of the loudspeaker together with the effects of spherical spreading, i.e. geometrical sound attenuation, and attenuation due to air absorption. Atmospheric

absorption is primary dependent on sound frequency and secondary on air temperature and humidity. The attenuation of sound level is about 8-9 dB for a distance of 100 m for the used sound frequency of 7 kHz and typical values of meteorological quantities (DIN ISO 9613-1, 1993, temperature: 15 °C, relative humidity: 50 %, air pressure: 101325 Pa). Together with spherical spreading, a sound attenuation of 49-55 dB results for distances between 50 and 70 m. This free-field attenuation is always occurring and must be considered if one prepares the amplifiers and loudspeakers for

measurements.

In practice, the sound source and the receiver are close to the ground which makes sound propagation more complex. There are not only direct sound waves between loudspeaker and microphone, but also ground-reflected sound waves (Fig. 7). The interference between those sound waves can lead to considerable effects which are estimated hereafter.

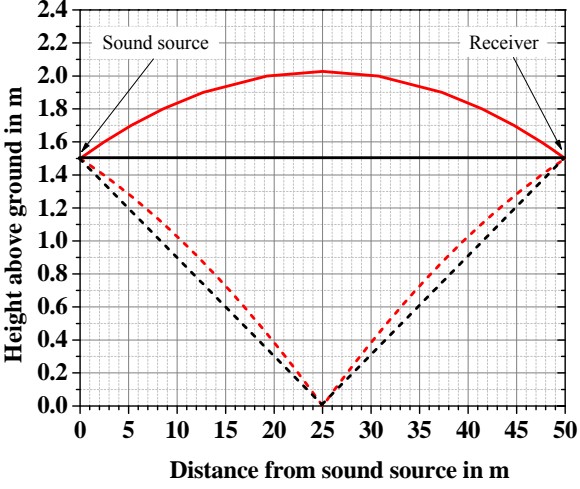

**Figure 7: Scheme of sound wave reflection at the ground surface: direct (solid lines) and reflected (dashed) sound paths, with (red) and without (black) atmospheric refraction due to sound speed gradients.**





Reflection at ground surface

To estimate the effect of reflection at ground surface, an idealized case is considered (see Ostashev and Wilson, 2016): the air and ground are homogeneous half spaces without any ambient motion. It follows, that the total sound field at a receiver may be assumed as the sum of sound traveling along a direct path from the source, plus sound traveling along a path that is

reflected by the ground (Fig. 7, black lines). Thereby, waves propagating along the air/ground interface are not included. It is reasonable to use this assumption so long as the angle between the ray path and the ground is not too small (nearly grazing sound incidence).

Assuming that the two sound waves are coherent, there is a constructive or destructive interference. The sound level of the received signal increases or decreases compared to the free-field, unbounded sound propagation. Calculations after Salomons

(2001) for a spherical sound wave travelling through a homogeneous atmosphere with reflection at a homogeneous ground surface are dependent on the sound propagation geometry (path length differences of the direct and the reflected path), the sound frequency, and the reflection coefficient. The latter one is influenced by the impedance of the ground surface which is usually parameterized by the sound frequency and the acoustic flow resistance (Delany and Bazley, 1970).

Commonly, the so-called relative sound level, i.e. the difference between the sound pressure level with and without (i.e.

unbounded free-field sound propagation) ground surface, is applied to quantify the ground effect at the receiver (Ostashev and Wilson, 2016). A positive relative sound level marks amplification (maximum of 6 dB), a negative one an attenuation of sound level (in theory, an infinitely high attenuation is possible).

It is essential for a high accuracy of acoustic travel-time measurements to provide SNR as large as possible at the receiver. Hence, a positive relative sound level should be ensured which can be realized using a suitable combination of sound

frequency, distance between loudspeaker and microphone as well as heights of the acoustic devices above ground surface. Values of relative sound level for a grassland site (with acoustic flow resistance of 200 kPa s$^{-1}$ m$^{-2}$) and the geometry of A-TOM measurements are shown in Fig. 8. For more detailed information to the calculation steps, please see e.g. Salomons (2001) and Ziemann et al. (2013).

For a distance of 50 m between loudspeaker and microphone and a signal frequency of 7 kHz, the relative sound level is near

or greater than 0 dB for both heights (Fig. 8a, b). That means an amplification of received sound level due to the ground effect. Higher or lower frequencies cause a so called ground dip, i.e. a strong decrease of sound level due to negative interference phenomenon. The greater the height of acoustic devices above ground surface, the higher is the sensitivity of the relative sound level to frequency (Fig. 8b in comparison to a). An increasing distance from sound source (50 m in comparison to 70.7 m, the latter one corresponds to the diagonals of A-TOM measurement field) mitigates the risk of a

ground dip in the investigated frequency range.





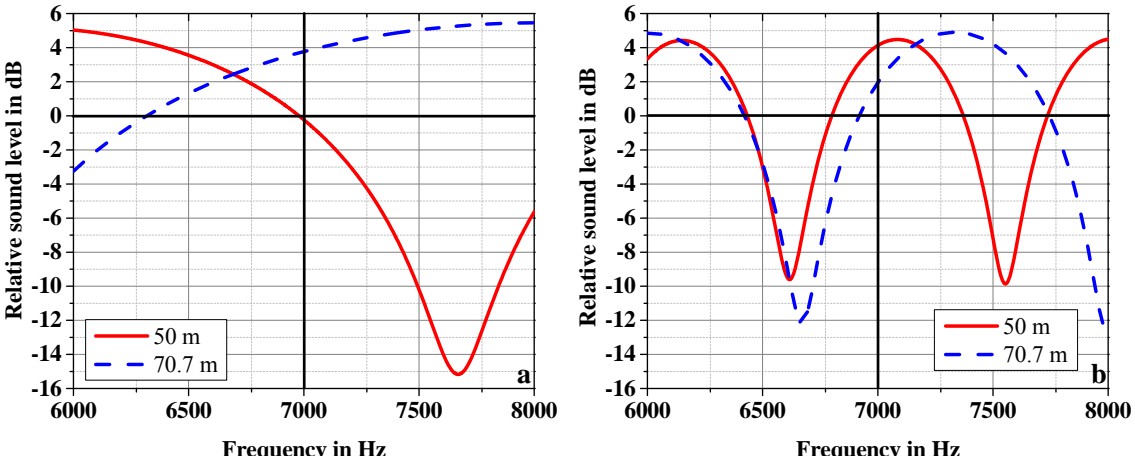

**Figure 8: Relative sound level depending on the sound frequency and on the distance (red solid line: 50 m, blue dashed line: 70.7 m) from the sound source to the receiver for a grassland surface. The height of the acoustic devices above ground is 1.5 m (a) and 3.0 m (b), respectively.**

5   Fig. 9a shows again the lower number of ground dips for the lower measurement level of 1.5 m above ground surface. For an increasing height of 3 m above surface (Fig. 9b), the sensitivity of relative sound level on the distance increases due to a growing number of ground dips. Furthermore, the sound level attenuation increases for a growing distance. Thus, sound path lengths of 50 m and 70 m together with a signal frequency of 7 kHz are favourable because of an optimized SNR of the received signal. Additionally, Fig. 8 and Fig. 9 demonstrate the requirements for the frequency stability of the used

10  loudspeakers.

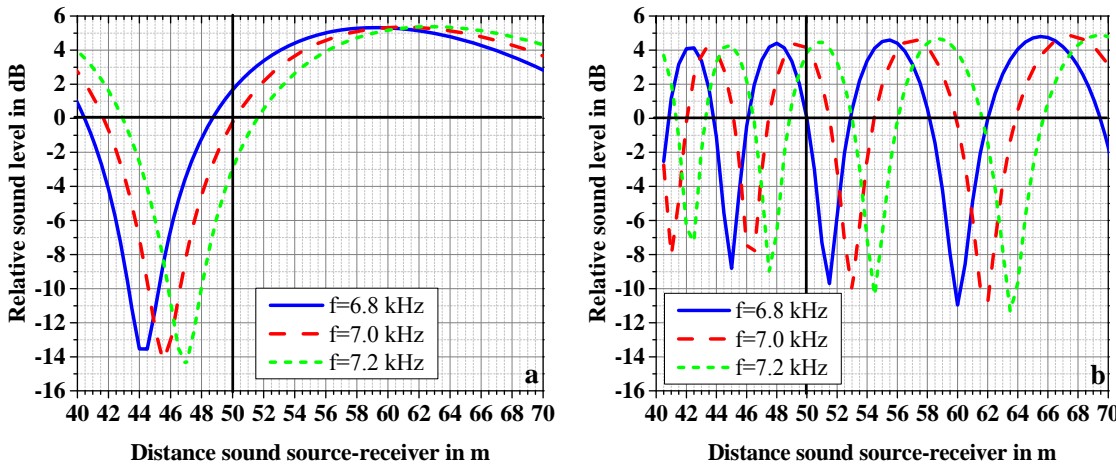

**Figure 9: Relative sound level depending on the distance and on the sound frequency for a grassland surface. The height of the acoustic devices above ground is 1.5 m (a) and 3.0 m (b), respectively.**




As already mentioned, the received signal at the microphone contains two parts. First one is the direct wave propagating through the air layer at one and the same height above ground between sound source and receiver. Second one is a wave that is reflected by the ground. This wavelet integrates the conditions of the air layer between the ground surface and the receiver. Assuming coherent waves and long enough signals emitted by the loudspeaker, the above described interference effects are

possible.

For outdoor sound propagation, atmospheric turbulence occurs and results in phase and amplitude fluctuations of the sound waves. This effect reduces the coherence between the direct and the reflected sound wave followed by partly attenuated and blurred interference impacts on the measured sound level. Especially the ground dip is reduced due to turbulence which increases the SNR at the receiver for special sound frequencies and propagation geometries. In this way, the results of Figs. 8

and 9 show rather extreme values of the ground effect influencing the received sound level without atmospheric turbulence, e.g. during nighttime conditions.

Additionally, the finite length of the signal (Fig. 5) has to be considered to evaluate the ground effect. It was examined whether the directly propagating and the reflected sound wave parts could be separated due to their time delay at the receiver. Thereby, straight-line sound paths, i.e. a homogeneous atmosphere, were again assumed. The time difference

between direct and reflected signal arrivals is growing up with increasing height above ground of acoustic devices (Table 3).

Table 3: Time difference (in sample units) between signal arrivals of direct and ground-reflected wave parts for a constant and homogeneous sound speed (temperature of 8 °C, calm).

| Distance source-receiver in m | Height above ground in m | |
|---|---|---|
| | 1.5 | 3.0 |
| 50 | 14 | 55 |
| 70 | 10 | 39 |

The greater the distance to the receiver, the smaller is the time difference. For the sound propagation at the lower level (1.5 m above ground) and a sound frequency of 7 kHz, i.e. period duration of about 0.14 milliseconds (approx. 7 sample units), the signals of direct and reflected waves cannot be distinguished because the signal itself has a length of approx. ten periods (approx. 1.4 ms = 72 sample units). This leads to a received signal containing the acoustic ground effect in the measured sound level. The strength of this effect depends on the amount of atmospheric turbulence and the interference of

direct and reflected sound waves. Furthermore, the received signal contains partially the properties of the atmospheric layer between ground surface and microphone. Hence, the real measurement height of the acoustically derived wind velocity and temperature can be slightly smaller than the geometrical height of the acoustic devices above ground. Using a shorter signal length, the onset of direct and reflected signals could be distinguished, but along with decreasing certainty of travel-time determination.





In addition to the effect of reflection at ground surface, refraction due to wind and temperature gradients has to be considered for outdoor sound propagation.

Refraction due to sound speed gradients

Atmospheric refraction can be described as a changed propagation direction of sound waves (e.g. Salomons, 2001). The resulting curved sound paths lead to a deviation from the straight-line sound propagation. The assumption of reciprocal sound propagation, i.e. along straight lines between transmitter and receiver, allows the simplified separation between the temperature and the wind influence on the acoustic travel time (Eq. (9)). However, it is questionable to what extent the refracting effect due to temperature and wind gradients affects this assumption. Thereby, vertical wind and temperature

gradients are especially important because they are usually greater than horizontal ones.

At first, the effect of downward refraction on the travel-time measurements is estimated because this kind of refraction happens usually during cloudless nights with a stably stratified atmosphere. Downward refraction occurs due to positive gradients of effective sound speed (see Eq. (3)), for instance during a temperature inversion or/and for a sound propagation in wind direction assuming an increasing wind speed with height above ground. If one supposes, that the curved rays can be

approximated by circular arcs (strictly speaking only valid in a motionless medium) depending on a constant vertical sound speed gradient in a stratified atmosphere (e.g. Attenborough et al., 2007), then the path length differences d$l$ between curved (first term) and straight-line ray (second term = $d$) can be calculated from Snell's law as follows:

$$d l = \sin\alpha_S \int_{z_S}^{z_R} \frac{c_{eff}(z)}{\sqrt{c_{effS}^2 - c_{eff}(z)^2 \sin^2\alpha_S}} \, dz - d \quad \text{with} \quad c_{eff}(z) = c_{effS} + z \frac{d c_{eff}(z)}{dz} \tag{20}$$

Here, $\alpha_S$ is the emission angle at sound source (polar angle of sound path measured from the positive z-axis, $\alpha_S > 0$), $z_S$ and

$z_R$ are the heights of source and receiver, respectively, $c_{effS}$ is the effective sound speed at the height of sound source and $d c_{eff}(z)/dz$ marks the constant vertical sound speed gradient. This equation can be easily solved in discretized form with finite thickness of several atmospheric layers. Thereby, the emission angle of sound rays is varied step by step until getting a connecting line between sound source and receiver point of the given measurement set-up. To estimate typical values of effective sound speed profiles on a cloudless summer day similar to experimental conditions, a numerical simulation of

meteorological conditions was performed using HIRVAC (HIgh Resolution Vegetation Atmosphere Coupler; Mix et al., 1994, Ziemann, 1998, Goldberg and Bernhofer, 2001). The two-dimensional version of this boundary layer model (approx. 100x100 model layers, Queck et al., 2015) is solving the basic equations for momentum, temperature and humidity. It contains additional terms describing the exchange of energy and humidity between vegetation and atmosphere at each model level. Calculation of temperature, wind velocity, and humidity profiles was followed by a calculation of the effective sound

speed in sound propagation direction and its vertical gradients as average over 30 min for several local times (Fig. 10).

At the transmitter height of 1.5 m or 3 m, positive gradients can be expected for a sound propagation in wind direction. In general, the gradients increase with decreasing height. Highest downwind gradients occur at nighttime and reach strong values of 0.57 s$^{-1}$ (0.25 s$^{-1}$) at a height of 1.5 m (3 m). In comparison, the gradients at noon are significantly smaller mainly



due to differences in the temperature profile between night (temperature inversion) and day (decreasing temperature with height). Figure 7 (red lines) shows an example for the calculated curved sound rays applying a sound speed gradient of about $0.6\ s^{-1}$. The height of the curved sound path above the measurement height of 1.5 m (3 m) is about 0.5 m (0.2 m) for a distance of 50 m and 1.0 m (0.5 m) for a 70 m distance between sound source and receiver. Over this height range, the direct sound path between loudspeaker and receiver integrates atmospheric conditions due to refraction. Additionally, the effect of ground reflection occurs again (Fig. 7, dashed red lines) which leads to a further integrating effect of a height layer with finite thickness around the measurement height.

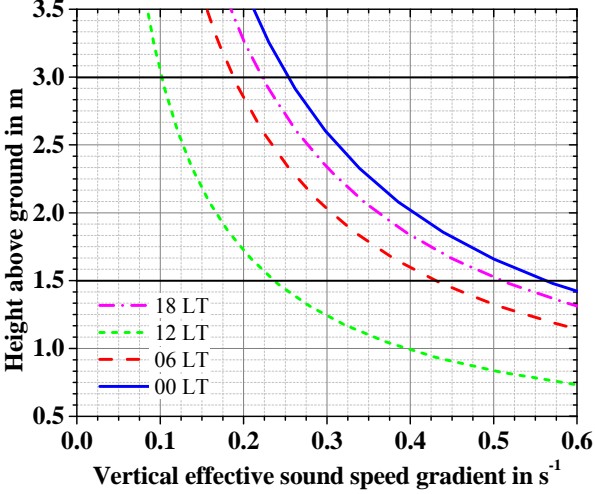

**Figure 10:** Vertical profiles of effective sound speed gradient (30-min-mean) in sound propagation direction simulated by HIRVAC for homogeneous grassland (vegetation height=0.3 m, Leaf Area Index=2) on 15th of July (exemplarily for a cloudless summer day similar to experimental conditions) for different daytimes (LT=Local Time).

Outgoing from the simulated vertical sound speed gradients, a travel-time difference between curved and straight-line direct sound path is calculated according to Eq. (20), including the different sound speed values along the different sound paths (Table 4). Please note that the used sound speed gradients are the maximum values in the simulated diurnal cycle. Therefore, the uncertainty estimation above represents a rather conservative estimation.

These travel-time differences are mostly smaller than the travel-time uncertainty due to the signal analysis (4 sample units, see Sect. 4.1.1). Especially for short distances at a height of 3 m the difference is negligible. Only at longer distances and smaller measurement heights above ground, the same magnitude of uncertainties occurs. In this case it has to be proven during the further data and uncertainty analysis, that the measured vertical sound speed gradients are similar to the simulated ones. Thus, considering downwind gradients especially for nighttime conditions, the vertical sound speed gradient should be controlled, e.g. using accompanying ultrasonic measurements to ensure the applicability of reciprocal sound propagation.



**Table 4: Comparison of travel-time uncertainties: Above: Travel-time difference (in sample units) between straight-line and curved sound path through the atmosphere for a maximum vertical gradient of effective sound speed of 0.6 s$^{-1}$ (during nighttime) on a summer day over grassland. Below: travel-time uncertainty due to signal analysis using (CCF), see Sect. 4.1.1.**

| Uncertainty due to travel-time difference between straight-line and curved sound path | Distance source-receiver in m | Height above ground in m | |
|---|---|---|---|
| | | 1.5 | 3.0 |
| | 50.0 | 2 | 0 |
| | 70.0 | 6 | 1 |
| Uncertainty due to signal analysis of travel time measurements | 50.0/70.0 | 4 | |

The analysis of measured vertical temperature gradients shows (see Sect. 4.3) that the above presented estimation of uncertainty mostly reflects a worst case. For further investigations in this study, the data at a height of 1.5 m above ground were used only for the short distance of 50 m. The deviation from the straight-ray approximation leads in this case to an additional travel-time uncertainty of 2 sample units according to Table 4.

Finally, the sound propagation against the wind direction is considered. Only negative sound speed gradients result from the investigations with the boundary layer model HIRVAC. Maximum gradients occur at midday (here not shown). This leads to an upward directed refraction of the sound waves in the atmosphere. For such conditions, no signal reaches the microphone which is located at one and the same height level as the loudspeaker. Nevertheless, due to a finite extent of the microphone, its spherical directional pattern and due to the scattering effect of atmospheric turbulence (Salomons, 2001), it is mostly possible to detect a signal in upwind direction if the wind speed and therewith the gradient is moderate. However, a low SNR frequently occurs in this case which has to be included in the travel-time analysis with adequate data quality flags. If a travel-time could be analyzed, the above explained uncertainty estimation for downward refraction can also be applied for the upward refracting case.

To sum up the outcomes of Sect. 4.1, following maximum uncertainties result for measurements at a height of 1.5 m above ground and for distances between loudspeaker and microphone of 50 m: (1) 4 sample units due to signal analysis, (2) 2 sample units due to sound refraction. The resulting travel-time uncertainty of 6 sample units can be recalculated into an uncertainty of about 0.4 K and 0.3 m s$^{-1}$ for instantaneous temperature and wind measurements (see Sect. 4.1.1). Applying averaging over, e.g., 30 minutes results in statistical uncertainties of about 0.04 K and 0.03 m s$^{-1}$.

### 4.2 Uncertainty of OP-FTIR-CO$_2$ measurements

Despite the great potential of OP-FTIR spectroscopic measurements, the technology is not commonly used for ground-based micrometeorological atmospheric monitoring due to the uncertainties in obtaining reliable information from the measured



spectra (Cieszczyk, 2014). The uncertainties for the retrieval of gas concentration from OP-FTIR spectra can be classified in (1) ambient environmental influences, (2) instrumental influences, and (3) data processing influences.

The environmental pressure and temperature variations during the measurements imply the main important inherent influence for infrared spectral data. Horrocks et al. (2001) demonstrated that especially temperature has a significant impact

on retrieval error and is an important parameter under consideration for subsequent data processing. The challenge to determine gas concentration using passive OP-FTIR under conditions with changing temperatures was described by Cieszczyk (2014).

Following Eq. (17), the main drawback and source for uncertainty in concentration determination processing from passive spectra obviously result from the dependency of signal amplitude from the difference between background temperature $T_B$

(thermal IR radiation) and target compound temperature $T_{air}$, which is assumed to be in thermal equilibrium with considered air volume. Usually, for passive OP-FTIR remote sensing this temperature difference is only a few Kelvins which affects an increasing error for the difference between spectral radiance of the background and the air (Polak et al., 1995, Harig et al., 2006). Using the approach proposed by Polak et al. (1995) the impact on transmission spectra can be analyzed by introducing a disturbed air temperature $T'$

$$T' = T_{air} + \varepsilon_T \tag{21}$$

where $\varepsilon_T$ is a given temperature error. This error leads to an erroneous spectral radiance of the air volume $\varepsilon_L$

$$\varepsilon_L = B(\nu, T') - B(\nu, T_{air}). \tag{22}$$

The disturbed transmission $TR'(\nu)$ is than given by

$$TR'(\nu) = \frac{TR(\nu) - \varepsilon_L/L_0}{1 - \varepsilon_L/L_0} \tag{23}$$

Using Eq. (14) the disturbed absorbance can be calculated using $TR'(\nu)$. Figure (11) shows the relative absorbance error $\Delta A/A$, which is directly related to the error of column density $\Delta(c_T \cdot d)/(c_T \cdot d)$, as a function of $\varepsilon_T$ for various temperature differences $(T_B - T_{air})$. As expected, the error for absorbance is increasing enormously for small temperature differences. Reasonable absorbance errors can be achieved for an absolute value of $\varepsilon_T$ smaller than 0.4 K.





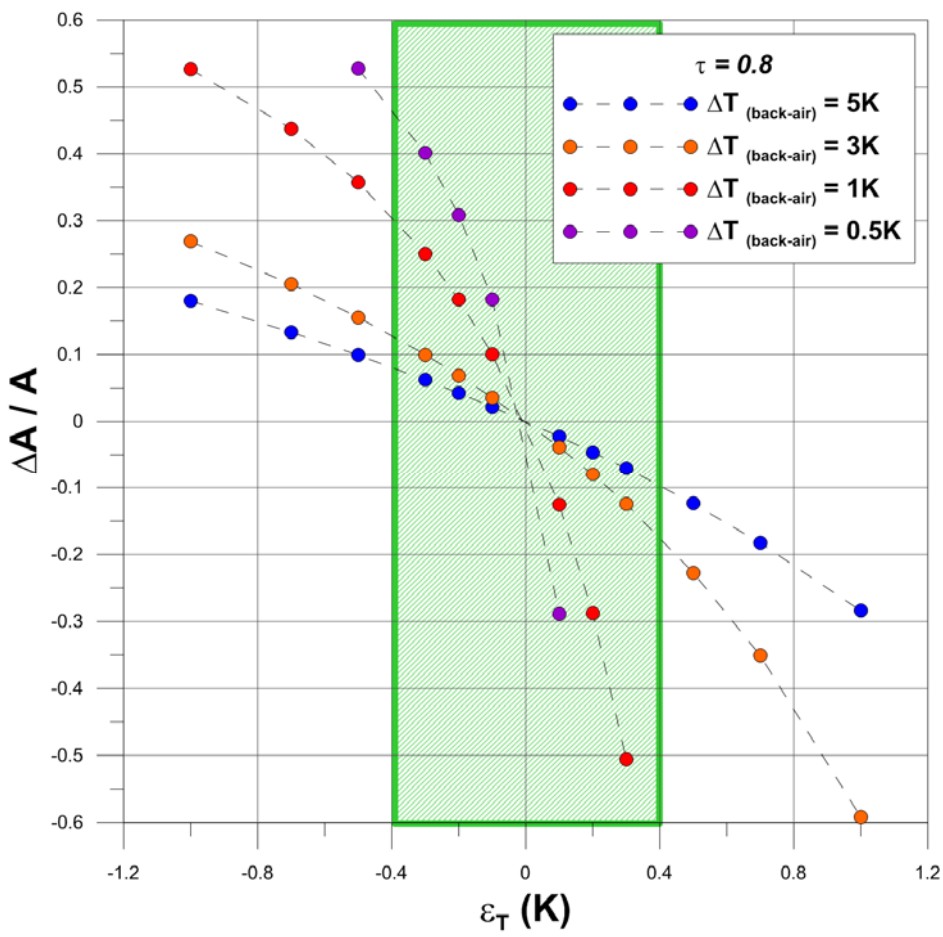

**Figure 11: The relative absorbance error $\Delta A/A$ as a function of a given temperature error $\varepsilon_T$ for various temperature differences $(T_B - T_{air})$. The errors were calculated for a transmission value $\tau = 0.8$ at wavenumber 800 cm$^{-1}$.**

5    In the case of Grillenburg experiment the passive radiance spectra were analyzed in accordance to Harig and Matz (2001) to determine the temperature difference between background and ambient air. In two spectral regions the spectra were fitted to the Planck radiation function using a non-linear least-square algorithm. In the spectral range less than 700 cm$^{-1}$, the atmosphere is more or less opaque and the spectral data contain the radiation temperature of the ambient air $T_{air}$ in the vicinity of the spectrometer device. The information on background radiation temperature $T_B$ was derived from the spectral

10    region between 850 and 1300 cm$^{-1}$. The obtained temperature differences $(T_B - T_{air})$ were compared to the horizontal temperature variability derived from 1-minute mean values of sonic temperature measurements, which is used as the presumed $\varepsilon_T$ (Fig. 12).





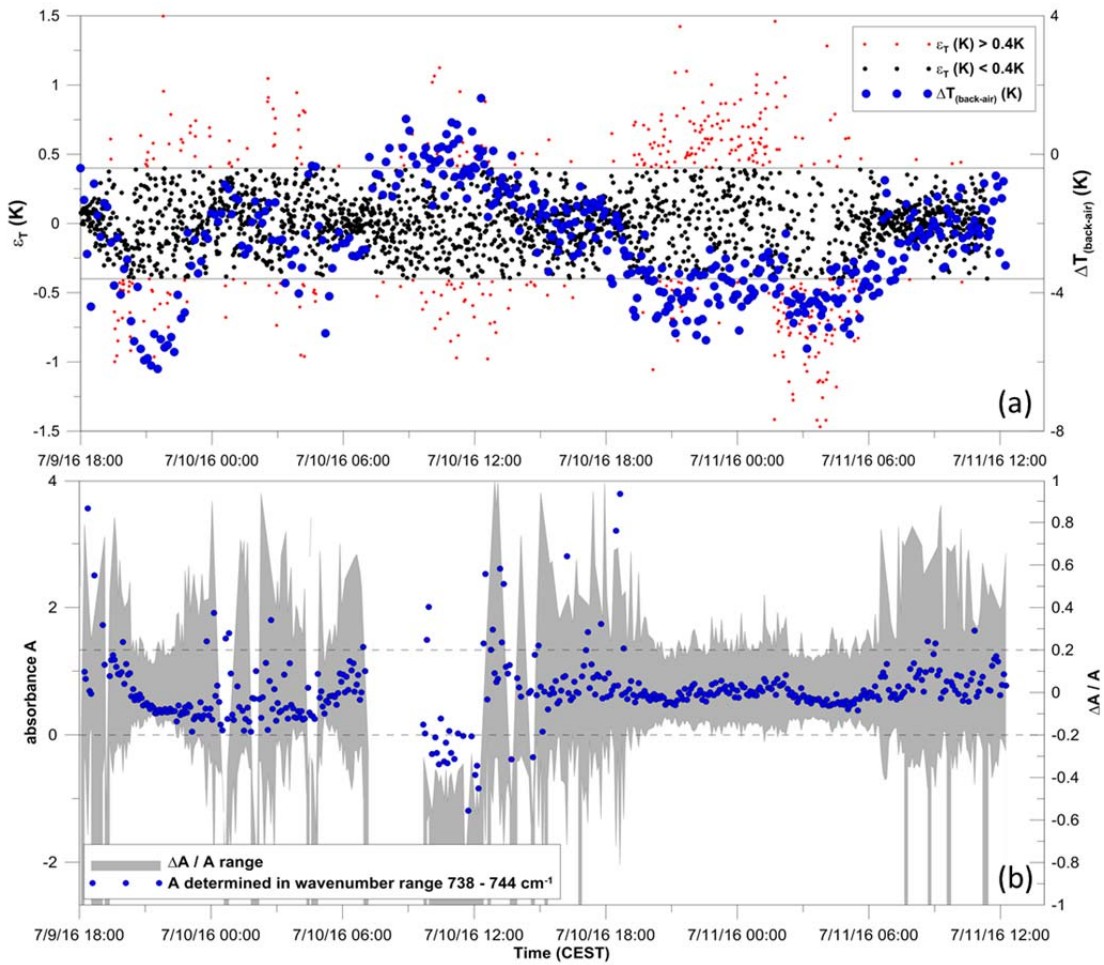

**Figure 12: (a) Comparison of obtained temperature differences ($T_B - T_{air}$) derived from passive radiance spectra and measured horizontal sonic temperature differences derived from two measurement points. Latter is used as estimation for the air temperature error $\varepsilon_T$. (b) The temperature data reveal measurements resulting in increased relative absorbance errors higher than 20% due to increased air temperature error $\varepsilon_T > 0.4$K and decreased temperature differences $|T_B - T_{air}| < 2$ K.**

For the considered period more than 90 % of the horizontal sonic temperature differences at two measurement points are less than 0.4 K. Furthermore, especially in the nighttime increased absolute values of temperature differences between background and air ($|T_B - T_{air}| > 2$ K) were observed. In these periods relative absorbance errors less than ±20 % are achievable. However, in periods around the noontime the passive radiance spectra reveal the thermal equilibrium of background and air. Hence, these periods are not suitable for further concentration analysis due to the extreme relative absorbance errors and have to be disregarded in the further data analysis.

From instrumental side the wavenumber resolution accuracy and the instrumental line shape (ILS) or apparatus function describes the influence of the spectrometer on the measured spectra. Each spectrometer device convolves the IR intensity




due to absorbance effects with this device characteristic function. The ILS is responsible for distortion of spectra caused by the finite detector area and finite optical path difference within the spectrometer. Most of the variation in ILS is driven by the instrumental resolution and the effective FOV due to misalignments of optical components inside the spectrometer. These doubts in true ILS of the applied spectrometer can lead to uncertainties in smoothing of spectral information and later on in concentration determination errors. Horrocks et al. (2001) estimated a concentration retrieval error of about 2 % due to an ILS uncertainty by measuring defined gas concentrations under fixed conditions. However, recent investigations concerning sensitivity of OP-FTIR retrievals by Smith et al. (2011) point out that using a broader spectral feature for concentration retrieval is suitable for the minimization of the effect of ILS on individual absorption lines.

The applied apodization functions (e.g., boxcar, triangular) and the internal optical path difference mainly control the influence in terms of spectral resolution. Manufacturer's maintenance specification concerning a wavenumber accuracy of 13 % at resolution of 4 cm$^{-1}$ was used to estimate an instrumental uncertainty based on simulation of absorbance spectra. The HITRAN Application Programming Interface (HAPI) is a set of Python routines for the easy access and processing of IR spectroscopic data for different gases and its isotopologues available in HITRAN database (Kochanov et al., 2016). The features of the modular routines provide, among others the receipt of the line-by-line data into a local database as well as the simulation of high-resolution spectra accounting for pressure, temperature, optical path length and instrumental settings. The influence of an uncertainty of wavenumber resolution on absorbance is shown as an example in Fig. 13. The simulation of the absorbance spectra includes environmental conditions similar to the Grillenburg experiment (T = 298 K, p = 1 atm, $CO_2$ line concentration = 40,000 ppm·m) and typical instrumental settings (e.g., triangular apodization function). The obtained relative absorbance errors $\Delta A/A$ range between 2 and 6.5 %.

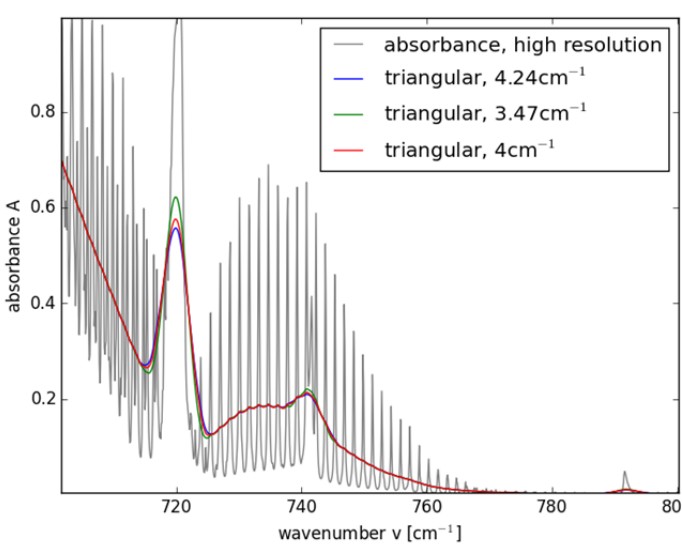

**Figure 13: Example of simulated $CO_2$ absorbance spectra (line concentration 40,000 ppm·m, T = 298 K, p = 1 atm) for wavenumber region 700 – 800 cm$^{-1}$. The relative absorbance error $\Delta A/A$ for a known uncertainty in wavenumber resolution accuracy given by manufacturer's specifications is in the range between 2 – 6.5 %. Besides the wavenumber accuracy, also the applied apodization function (here triangular) affects the relative absorbance error.**



Based on the previous data evaluation the absorbance spectra of the night time period from 10th – 11th July showed reasonable errors and were chosen for the subsequent quantitative analysis (Fig. 12b). This period covered an interval of 9.5 hours and included 108 spectra for each measured optical path direction. The concentration retrieval is based on chemometric techniques applied to the absorbance spectra deriving spectral properties which are related to quantitative

information. It included the usage of least-squares fitting comparing parts of the measured absorbance spectra with simulated reference spectra. The algorithm is previously well described for instance by Griffiths and de Haseth (2007) and Smith et al. (2011). Reference IR spectra including instrumental line shape were generated by using the HAPI routines (Kochanov et al., 2016). Additional Python routines were designed for the selection of spectral windows and the comparison of measured and simulated spectra based on the classical least-squares approach (CLS) as a straightforward algorithm (Shao et al., 2010).

Actually, different retrieval methods to obtain concentration values from measured spectra are available (e.g., CLS, partial least-squares regression PLS). Smith et al. (2011) observed an increasing underestimation of the CLS-based method at higher path lengths. However, for Grillenburg experimental setup the optical path lengths and the expected line concentrations were sufficiently low to use a CLS-based retrieval approach neglecting the Beer-Lambert law nonlinearity.

A spectral window ranging from 700 to 760 cm$^{-1}$ was used for the determination of $CO_2$ line concentrations due to the

significant absorbance feature of $CO_2$ molecules within this wavenumber region. The quantitative accuracy was determined from fit residuals of the calculated and measured absorbance spectra. Only measurements with valid fitting errors smaller than 3 % were defined as acceptable for further data analysis.

For the Grillenburg experiment the maximum uncertainty for $CO_2$ concentration determination from passive OP-FTIR measurements was estimated based on the considered systematic influences due to environmental parameters, instrumental

characteristics, and retrieval procedure with a total amount of approx. 30 % for a single measurement. This value for total uncertainty seems to be high compared to active OP-FTIR investigations (Horrocks et al. 2001, Smith et al. 2011). The uncertainty of temperature difference between background and considered atmospheric gas compound could be identified as the main error source for the passive measurements and a threshold of 2 K for data filtering was defined. In summary, the total uncertainty represents the maximum error estimation, valuable for the validation of the method in terms of applicability

to determine spatial concentration variations for the micrometeorological investigations addressed by this study. The estimated range of maximum concentration uncertainty for our experiment was confirmed by other passive OP-FTIR investigations (e.g., Allard et al., 2005, Sulub and Small, 2007, Kira et al., 2015). However, most of these studies are based on hot gases with high temperature contrasts between background and target gas compounds (volcanic gases, exhaust gases) or on the determination of non-atmospheric GHG gases (industrial gases, aerosols).

**4.3 Applicability of combined A-TOM and OP-FTIR measurements**

At this point, the uncertainties of the two methods, A-TOM and OP-FTIR, are known. Single, instantaneous values of wind components, measured by A-TOM, can be derived with an uncertainty of 0.3 m s$^{-1}$ for the described setup of the Grillenburg





experiment (height of 1.5 m above ground and path lengths of 50 m). After averaging over a time period of 30 min the statistical uncertainty amounts to 0.02 m s$^{-1}$.

The wind component in $x$-direction, $u$, was calculated as a spatial mean along the two paths between the measurement positions ATOM1 and ATOM4 as well as ATOM2 and ATOM3 (see Fig. 2). The sound path between ATOM1 and ATOM4

5    is parallel to the optical path R72 – R73 of OP-FTIR measurements. The perpendicular wind component $v$ was derived by averaging the line-integrated wind measurements between ATOM1 and ATOM2 as well as ATOM4 and ATOM3 (parallel to OP-FTIR-path R72 – Black4).

Figure 14 shows that the wind speed was relatively small during the exemplarily investigated nighttime in July 2016 at the Grillenburg site. Furthermore, after midnight the wind speed was steadily falling to mean values smaller than 1 m s$^{-1}$.

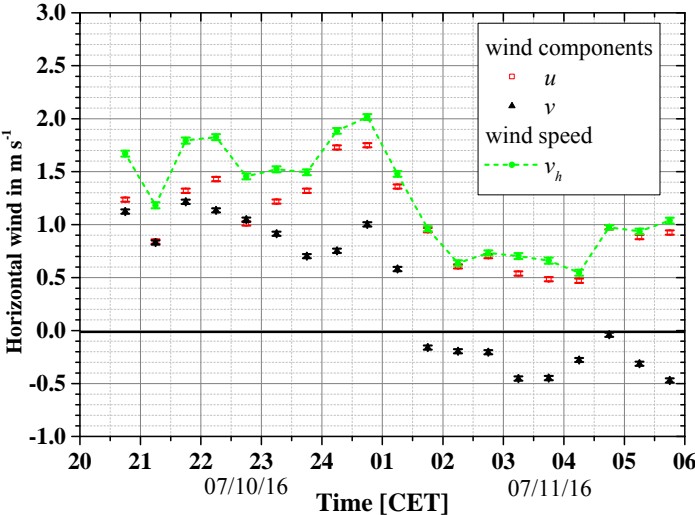

**Figure 14: Averaged (30 min) horizontal wind speed at a height of 1.5 m measured by A-TOM and with maximum uncertainties.**

These low wind conditions near the surface during a clear night were supported by a stably stratified atmosphere. Figure 15 determines a positive vertical temperature gradient during all nighttime hours.

15    Between 3 and 5 o'clock a noticeably high value of the temperature gradient occurs together with very small wind speed values and a changing wind direction shortly before the onset of this sharp increase in stability. Thereby, the A-TOM measurements are showing a similar behaviour in comparison to the measurements using sonic anemometers. Mostly, the spatially averaged data are similar to all point data. However, there are greater differences between the data from sonic anemometers especially during times of high vertical gradients and times of highly variable gradients, respectively.





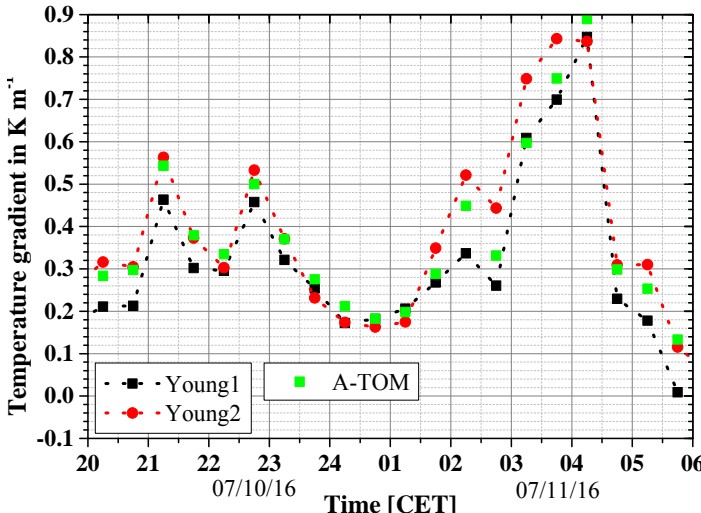

**Figure 15: Averaged (30 min) vertical gradient (3 – 1.5 m) of acoustic virtual temperature measured by sonic anemometers (Young1/2 see Fig. 2) and by A-TOM as spatial mean (50 x 50 m²).**

Absolute $CO_2$ concentrations, measured by OP-FTIR, are estimated with a maximum uncertainty of 30 % for a single measurement. Considering the application of averaging over a period of 30 minutes, the standard error of the mean values can be decreased at least by a factor $1/\sqrt{4} = 0.5$, because not all single data values could be used for further analysis. Recognising the time-dependent (i.e. concentration-dependent) calculation of uncertainty in Sect. 4.2, the single measurement uncertainty amounts to a maximum value between 20 and 30 %. Based on the recalculation of relative into an

absolute error values including the averaging time of 30 min and only considering all nighttime measurements, an averaged, statistical uncertainty of approx. 70 ppm is resulting. Smaller values of uncertainty can be obtained for smaller concentrations. Values with a determined uncertainty greater than 30 % are excluded from further analysis. Thereby, exemplary concentrations along the two paths between the measurement positions (see Fig. 2) R72 - Black4 (distance: 100 m) and R72 – R73 (80 m) were analysed. Figure 16 shows the temporal and spatial differences of $CO_2$ concentrations

along the two mentioned optical paths at the Grillenburg site during nighttime measurements. Again, the special time period around 4 o'clock is standing out with comparably higher concentrations accompanied by significant spatial differences.

The temporal and spatial variability of $CO_2$ concentration determined by OP-FTIR was compared to the results of the measurements taken by EC station (3 m above ground) and soil respiration chamber measurements at ground surface (Figure 17). Obviously, a distinct similarity in concentration time series is observable for all measurements. The point measurements

underlined a distinct variability in horizontal as well as in vertical distribution, also perceptible in OP-FTIR data. Furthermore, the chamber measurements at ground surface illustrated the increased spatial variability of $CO_2$ concentration during nighttime caused by soil respiration processes. The data of Grillenburg experiment clearly demonstrated the main





difference between line and point measurements: OP-FTIR measurements provided path-integrated values including the assumed spatial concentration variability and yielded spatially averaged concentration values.

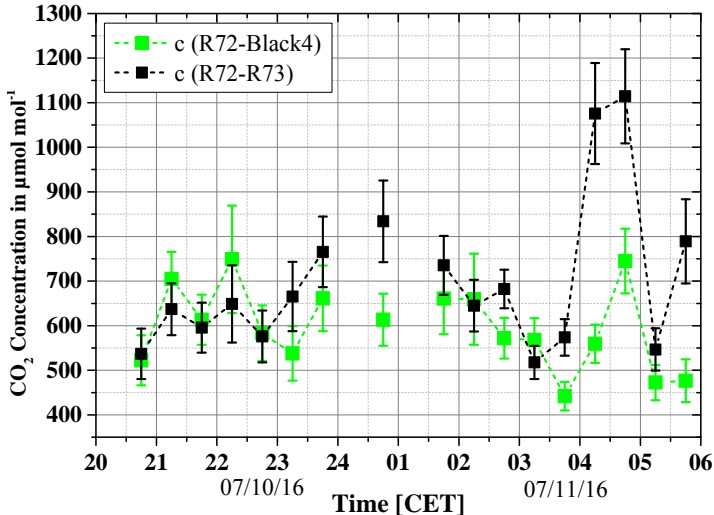

**Figure 16:** Averaged (30 min) $CO_2$ concentration measured by OP-FTIR at perpendicular paths at a height of 1.5 m above ground, representative for total investigation area with vertical extent due to field of view of 0.25 m, and with maximum uncertainties.

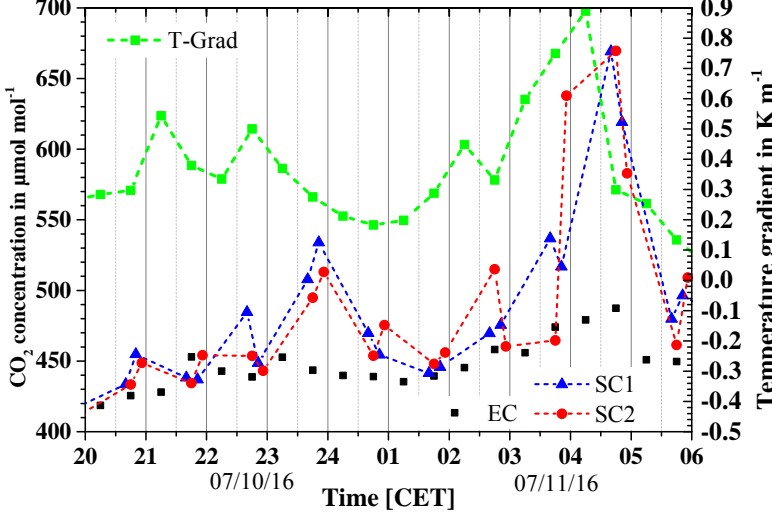

**Figure 17:** Averaged (30 min) $CO_2$ concentration measured by EC station at a height of 3 m, measured by two soil respiration chambers at the ground surface (SC1/2: horizontal distance between the chambers 5 m), and vertical temperature gradient measured by A-TOM (3 – 1.5 m).





In the next step of analyses, the horizontal advection and its uncertainty were calculated. Thereby, an adapted form of Eq. (2) was applied according to the analysed results so far:

$$F_{Hor} \approx \frac{1}{V_m} \left( \overline{v_h} \frac{\Delta \bar{c}}{\Delta d} \right) \Delta z \tag{24}$$

To roughly estimate the spatial concentration differences within the investigated area inside the square R72-Black4-Black2-R73, two line-integrated concentrations and their difference were used: R72 - Black4 and R72 − R73. Because these two paths are perpendicular and include the total acoustic measurement area, the horizontal wind speed $v_h$ was used in Eq. (24) instead of the wind components $u$ and $v$. In this way, Eq. (24) gives an estimation of the spatially averaged and representative horizontal advection at the Grillenburg site.

To derive the maximum uncertainty of horizontal advection at a certain height level above ground, the error propagation law is then applied to Eq. (24) with $\Delta d$ = 100 m for an averaged difference of distance:

$$\Delta F_{Hor} = \frac{1}{V_m} \left( \left| \frac{\Delta \bar{c}}{\Delta d} \right| \Delta \overline{v_h} + |\overline{v_h}| \frac{\Delta \Delta \bar{c}}{\Delta d} \right) \Delta z = \frac{1}{V_m} \left( \left| \frac{\Delta \bar{c}}{\Delta d} \right| \left( \left| \frac{\bar{u}}{\overline{v_h}} \right| \Delta \bar{u} + \left| \frac{\bar{v}}{\overline{v_h}} \right| \Delta \bar{v} \right) + |\overline{v_h}| \frac{\Delta \Delta \bar{c}}{\Delta d} \right) \Delta z. \tag{25}$$

Thereby it is assumed that the uncertainty of path length ($\Delta d$) estimations and layer thickness ($\Delta z$) determination is negligible in comparison to the uncertainties of wind components and spatial concentration differences. It has to be noticed, that the concentration error for a measurement along one optical path counts twice in the term $\Delta \Delta \bar{c}$ due to the spatial difference of concentrations $\Delta \bar{c}$. This behaviour results in relatively large values of the last term in Eq. (25), at least one magnitude larger than the first term which accounts for the wind uncertainty.

A value of $22.414 \cdot 10^{-3}$ m$^3$ mol$^{-1}$ was applied for the molar volume of dry air $V_m$. The vertical layer thickness $\Delta z$ is mainly influenced by the field of view of the OP-FTIR measurements. Assuming 10 mrad (see Table 1), an averaged vertical layer of 0.25 m is investigated along a path length of 100 m. Using these values together with Eq. (25) maximum uncertainties of $3 - 38$ µmol m$^{-2}$ s$^{-1}$ follow. In the light of temporary great values of horizontal advection and including the spatially averaging and expanding effects of the method, the uncertainties are reasonable. Nevertheless, there are several possibilities for further development of the combined method which will be discussed in Sect. 5.

Figure 18 shows the resulting estimation of horizontal advection at a height of 1.5 m above ground, representative for the total investigation area of approx. 120 m x 120 m and an vertical extent of 0.25 m due to the field of view of optical measurements. The spatial gradient derived from the spatial difference of $CO_2$ concentrations is the factor which decides the sign of advection because the wind speed is always positive. In this way, the sign of advection is a bit arbitrary.

The temporal behaviour of advection is generally connected with that one of spatial concentration difference, but it is modulated by the wind speed. Mostly, the temporal variability of advection is coupled with temperature gradient until 3 o'clock. During this first time period, the course of advection and atmospheric stability is similar: increasing stability occurs together with increasing advection and vice versa. The turbulent $CO_2$ flux demonstrates frequently a similar behaviour. During the following time period, the wind turns, wind speed decreases, atmospheric stability increases remarkably, and EC flux increases too. It has to be noticed that EC fluxes during such low wind conditions should be treated





with high caution (e.g. Aubinet et al., 2012). In comparison to that, the advection decreases sharply after 4 o'clock. This event is coupled with the rising near-surface concentration of $CO_2$ measured by the soil respiration chambers and to lower extent by the EC system (Fig. 17) shortly after reaching the maximum of temperature gradient.

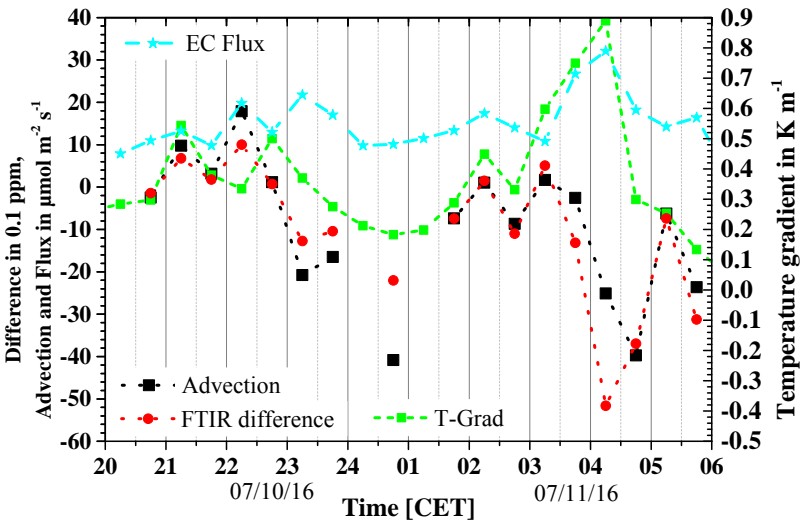

**Figure 18:** Averaged (30 min) spatial difference of $CO_2$ concentration (OP-FTIR) and horizontal advection (A-TOM, OP-FTIR) at a height of 1.5 m, vertical gradient (3 – 1.5 m) of acoustic virtual temperature (A-TOM) and $CO_2$ flux at a height of 3 m (EC).

## 5 Conclusions and outlook

To close the known gap within energy balance which affects the $CO_2$ balance determinations, there is still a considerable requirement for adequate advection measurements. Up to now, there are a lot of measurements approximating the required quantities between points at selected transects. It has been shown that especially more detailed spatial information about flow properties and $CO_2$ distribution in the control volume would be necessary (Feigenwinter et al., 2008). Ground-based remote sensing techniques can provide spatially representative $CO_2$ concentration values together with wind components within the same voxel structure. For this purpose, the presented SQuAd-approach applies an integrated method combination of line-averaging acoustic tomography to measure wind components together with open-path Fourier transform infrared spectroscopy to derive spatially integrated $CO_2$ concentrations.

The derived values of mean advection around 10 µmol m$^{-2}$ s$^{-1}$ (and sometimes higher) seem to be comparatively high (e.g. Zeri et al, 2010). Similar values of about 50 µmol m$^{-2}$ s$^{-1}$ for advection as well as $CO_2$ gradients of 1 µmol m$^{-2}$ s$^{-1}$ were detected in more complex environments regarding topography and vegetation cover (Feigenwinter et al., 2010). In this respect our results at relatively flat grassland site maybe worth looking into. Thereby, the different measurement volumes of point-like and line-averaging measurement methods should be taken into account. It is expectable that spatially integrating and representative measurements of concentrations lead to higher values in comparison to point measurements which could



be affected by undersampling of real-world fluxes (Siebicke et al., 2011). Furthermore, the shown exemplary results were measured near the ground during stable stratification with remarkable amounts of temperature gradient as well as during low wind conditions. Several authors, e.g. Sun et al. (2007), Kutsch et al. (2008), Siebicke et al. (2012), found maximum advection during such conditions especially near the ground surface (Feigenwinter et al., 2008). The analysis of further data
sets with additional concentration measurements and for additional time periods should confirm the derived results so far and the possibility to apply spatially averaging methods to measure advection of $CO_2$.

To demonstrate the applicability of the SQuAd-approach, the estimation of uncertainties of the used measurement and analysis methods was in the focus of attention. Thereby, it is important to notice that we applied a maximum error calculation of the used methods A-TOM and passive OP-FTIR to be on a safe side for further applications. The received
values of uncertainties (0.3 m s$^{-1}$ for wind components and 30 % for concentration for instantaneous data without averaging) are always greater in comparison to an investigation of purely statistical uncertainties, i.e. random errors which are usually described by the standard deviations of high-frequency measurements (e.g. Marcolla et al., 2014).

Nevertheless, there are still possibilities to further decrease these uncertainties. Thereby, the data analysis of $CO_2$ concentrations will focus on all other optical paths of the passive OP-FTIR measurements as well as on the active OP-FTIR
data. The generated data redundancy will allow enhancing the security of measurement results. In this way, the presented estimation of maximum uncertainty will be reduced to smaller values which are typical for micrometeorological applications. Additionally, a higher frequency of measurements would decrease the statistical uncertainty of both methods, A-TOM and passive OP-FTIR.

Further tests to improve the accuracy of the applied OP-FTIR method will focus on an increasing temperature gradient
between background and target gas as well as the determination of the influence of FOV on horizontal and vertical resolution. The integral concentration value based on spectral information along the optical path includes a smearing effect caused by the true FOV. Especially, for longer pathways and increased horizontal concentration gradients this effect has to be taken into account. Furthermore, slight misalignments can result in decreased data quality due to an unpredictable uncertainty in effectively considered path lengths and background radiations.

At the expense of temporal resolution and assuming stronger concentration differences between background and the searched air volume, the spatial resolution of the OP-FTIR-method can be further enhanced by measuring along a higher number of paths. In a similar way it is possible to increase the number of acoustic paths through the control volume. Thus, a highly enough number of optical and acoustic paths results which can be used to apply a tomographic algorithm and to reconstruct spatially resolved wind and concentration fields.

The presented SQuAd-approach offers the possibility to complement previous findings of multi-location, point-like measurements. Thomas (2011) found fundamental differences in the space–time structure of the motions dominating the variability of the wind and temperature fields. This scale mismatch complicates the derivation of meaningful estimates of horizontal advective fluxes without dense spatial information. The SQuAd-approach could be applied to provide the





necessary spatially representative data. Thereby, one advantage of the A-TOM/OP-FTIR-method is the combined measurement of wind components and temperature together with several GHG's along similar paths and air volumes. Though, there are still tasks concerning the improvement of combined measurement methods within the SQuAd-approach, the present study provides first examples of applying the new method to estimate a spatially representative advection during

calm and stably stratified nighttime conditions at a grassland site.

## Appendix A

Using the assumption of reciprocal sound propagation (see Eqs. (9) and (10)), it follows for the uncertainty of the acoustic virtual temperature $\Delta T_{av}$ and wind component along sound path $\Delta v_{Ray}$:

$$\Delta T_{av} = \left(\left|\frac{\partial T_{av}}{\partial d}\right|\Delta d\right) + \left(\left|\frac{\partial T_{av}}{\partial \tau_{forth}}\right|\Delta \tau_{forth}\right) + \left(\left|\frac{\partial T_{av}}{\partial \tau_{back}}\right|\Delta \tau_{back}\right) \quad \text{and} \tag{26}$$

$$\Delta v_{Ray} = \left(\left|\frac{\partial v_{Ray}}{\partial d}\right|\Delta d\right) + \left(\left|\frac{\partial v_{Ray}}{\partial \tau_{forth}}\right|\Delta \tau_{forth}\right) + \left(\left|\frac{\partial v_{Ray}}{\partial \tau_{back}}\right|\Delta \tau_{back}\right). \tag{27}$$

Differential measurements outgoing from a known initial state increase the accuracy because errors of the path length measurement can be compensated. With $\Delta d = 0$ it follows from Eqs. (26) and (10):

$$\Delta T_{av} = \left(\left|\frac{\partial T_{av}}{\partial \tau_{forth}}\right|\Delta \tau_{forth}\right) + \left(\left|\frac{\partial T_{av}}{\partial \tau_{back}}\right|\Delta \tau_{back}\right) = \frac{1}{\gamma_{tr}R_{tr}}\frac{d^2}{2}\left(\frac{1}{\tau_{forth}} + \frac{1}{\tau_{back}}\right)\left(\frac{1}{\tau_{forth}^2}\Delta \tau_{forth} + \frac{1}{\tau_{back}^2}\Delta \tau_{back}\right) \tag{28}$$

Assuming that travel-time errors along one and the same path in opposite directions (forth and back) are identical to $\Delta \tau$, the

temperature uncertainty from Eq. (27) can be written:

$$\Delta T_{av} = \frac{1}{\gamma_{tr}R_{tr}}\frac{d^2}{2}\Delta \tau \left(\frac{2\sqrt{\gamma_{tr}R_{tr}T_{av}}}{d}\right)\left(\frac{2(\gamma_{tr}R_{tr}T_{av})+2v_{Ray}^2}{d^2}\right). \tag{29}$$

The uncertainty of relative wind measurements is only depending on the uncertainty of travel-time measurements:

$$\Delta v_{Ray} = \left(\left|\frac{\partial v_{Ray}}{\partial \tau_{forth}}\right|\Delta \tau_{forth}\right) + \left(\left|\frac{\partial v_{Ray}}{\partial \tau_{back}}\right|\Delta \tau_{back}\right) = \frac{d}{2}\left(\frac{1}{\tau_{forth}^2}\Delta \tau_{forth} + \frac{1}{\tau_{back}^2}\Delta \tau_{back}\right). \tag{30}$$

## Acknowledgements

At first we want to thank our project partner Christian Bernhofer (Chair of Meteorology, TU Dresden) for initial impulses for the project, providing staff and the equipped experimental site Grillenburg. We sincerely thank Armin Raabe (Leipzig Institute for Meteorology, University of Leipzig) for quick and easy loan of acoustic devices for A-TOM. Many thanks are going to Markus Hehn, Valeri Goldberg, Uwe Eichelmann, Heiko Prasse (Chair of Meteorology, TU Dresden), Andreas Schoßland and Uta Sauer (Helmholtz Centre for Environmental Research Leipzig) for their support during preparation and

implementation of the experiment.
This work was supported by the German Research Foundation (DFG) [grant numbers ZI 623/10-1, SCHU 1428/3-1].




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
