# Peer review of "Line-averaging measurement methods to estimate the gap in the CO2 balance closure – possibilities, challenges and uncertainties"

_Atmospheric Measurement Techniques, 2017_

## Referee Comment (RC1) · Anonymous Referee #2 · 26 Jun 2017

General Summary and Comments: Overall the paper is well structured and detailed, specially through Sections 2-4. Combining some of the figures into one group of figures would be useful (e.g. Figures 8 and 9 or Figures 14 to 18) since they are often talked about concurrently. It would also be beneficial to combine or summarize the uncertainties shown in Tables 2-4 with respect to the temperature and wind values for better readability. The results sections are light on details why the CO2 concentrations for the proposed approach are as different compared to the EC measurement (Specific Comments). Another sentence or two pointing toward the potential error sources and how it could be reduced would be useful for further studies. Overall well put together, but some minor revisions necessary.

There are spots where the writing needs to be cleaned up and clarification but these do not impede the reading of the work (See Technical Comments below).

Specific Comments:

Pg. 5, Ln 5-7: Are there not effects from wind passing from behind the microphones on the sound collection or are the microphones directional so this is not a concern?

Pg. 9, Ln 17. What was the output power of the speakers?

Pg. 12, Ln 13-17: Can you provide more detail on the soil chamber measurement cycle? How long were the chambers closed for each interval and how was the concentration measured (see comment for Pg. 33, Figure 17).

Pg. 13, Ln 5: "growing sound frequencies"; what does this mean with respect to the 7-kHz frequency used?

Pg. 18, Ln 5-6: How much does the uncertainty decrease with the increased path lengths? If this is the case, why not use as long of a path length as possible instead of the "minimal path length of 50m"? What is the minimum path length that would generate usable data?

Pg 18. Ln 22: Section 3.2.1 didn't mention any amplifiers. Were amplifiers used and how did you generate the sound wave for the speakers?

Pg. 20, Ln 20-21: How stable was the frequency of the loudspeakers? How much of an error would this contribute to the results or was it negligible?

Table 2: Could the corresponding wind and temperature errors be added to this table and Tables 3 and 4?

Pg. 21, Ln 14-15: Overnight turbulence is not zero, there is still some minimal turbulence, but it may be negligible for this application. How much does turbulence negate the ground effects? Would this mean that daytime measurements are less error prone so better to validate the results of this method? AMTD
Pg. 27, Ln 6-8: Are the sonic temperatures the virtual sonic temperature or converted to ambient air temperature? The text refers to Tair and sonic temperature ( $\theta v$ ).

Pg. 28, Ln 3: Why was 20% chosen as the error level used?

Pg. 29, Ln 8: What constitutes "reasonable errors"?

Pg. 32, Figure 16: The FTIR CO2 concentrations are high compared to the EC measurements (Figure 17); is this a product of the line averaging or error within the FTIR/EC measurement systems?

Pg. 33, Figure 17: Soil chambers and EC CO2 measurements are not equivalent due to the height difference thus proximity to the source and the measurement style. At what point in the soil chamber cycle did the concentration measurement take place? Could you combine Figures 16 and 17 so the comparison between the concentration values were easier?

Pg. 33, Ln 7-8: Is the difference a product of the potential error within the measurement technique? In Ln 13-15 on Page 30 you state that the uncertainty was determined in studies based on with stronger temperature contrasts between the target gas and ambient air so couldn't the maximum uncertainty be even larger here? With only one point measurement of ambient CO2 (EC station), I don't think it is reasonable to make a generalization like this without support from other works. There was a 100+ ppm difference in the line average (R72-R73) compared to the point measurement (EC), only approximately 10 m away; that large of a difference in the concentration needs to be explained.

Pg. 34, Ln 12: The range of uncertainty is very large and could change the sign of the horizontal advection values. Does the error range scale with the magnitude of the advection? How do we account for the uncertainty when comparing with EC-based advection?

Pg. 34, Ln 18: What was the wind direction for this period? Did it remain constant or

AMTD
did it change with time? The change in sign of the v component in Figure 14 implies a changing wind direction and does this change the upstream source region for CO2 hence much larger advection?

Pg. 35, Ln 24-26: "It is expectable that...". My understanding is the differences in concentration measurements is a product of measurement height and measurement principle, not just line versus point measurements. Some different is expected from the line versus point measurement but there were points where the line average concentration (FTIR) doubled the point concentration (EC and SC) (Figures 16 and 17). This magnitude of difference in the concentration values doesn't seem to be just a difference in line versus point measurement averaging. I would expect the SC concentrations to be highest because of their proximity to the ground (CO2 source). Under-sampling of fluxes is different than under- or over-sampling the concentration values and the resulting advection values.

Pg. 35, Ln 20: What is the error on the cited horizontal advection values; is the range you found similar to these studies?

**Technical Comments**

Pg. 1, Ln 22: "Thereby...", "Additionally" may be a better word.

Pg. 1, Ln 27: Considering averaging..." does not need "Considering" and is missing a comma: "Averaging over a period of 30 minutes, the standard error".

Pg. 2, Ln 2: "A closing gap for balance..." would be better as "Closing the gap for..."

Pg. 2, Ln 4: "measurement of flows" awkward phrasing

Pg. 2, Ln 31: "considered advective fluxes" Vertical or horizontal advection?

Pg. 3, Ln 1: This sentence is misleading since substantial advection can occur in any land-cover type, the only requirement is a CO2 gradient to exist (e.g. Feigenwinter et al., 2008). I believe what the authors are attempting to convey is advection is commonly

AMTD
the largest error source in complex terrains.

- Pg. 3, Ln 5-6: How much reduction in annual CO2 update? And in which forest?
- Pg. 3, Ln 27: Remove extra parenthesis
- Pg. 5, Ln 19: Replace "several" with "two".
- Pg. 6, Ln 25: Don't need "Then,".
- Pg. 6, Ln 29: Not a complete sentence; rewrite.

Pg. 7, Ln 18-19: Combine "The permanent EC..." and "Meanwhile is it..." into one sentence. Define "ICOS-D (C3 station)".

- Pg. 8, Ln 10: "class-a-pan" should be "Class A pan".
- Pg. 9, Ln 17: "speakers for frequencies" should be "speakers with frequencies".
- Pg. 14, Figure 6: Could you add the sent signal as well to illustrate the time shift?
- Pg. 17, Ln 26: Need a comma between "50 m" and "a maximal".
- Pg. 18. Ln 4: Remove "For", so "A minimal path length...".

Pg. 19, Ln 16: "The latter is influenced...", don't need "one".

Pg. 21, Ln 5-10: This paragraph feels out of place. Some of this information was presented earlier in the section.

Pg. 21, Ln 16-17: "It was proven, whether...", which of the two were proven, that you can separate the two sound wave parts or it wasn't possible to separate the two parts?

Pg. 22, Ln 9-10: Repeat from earlier.

Pg. 22, Ln 22-23: "...vertical wind and temperature gradients..."; vertical gradient of the horizontal wind? And what "horizontal ones"? Horizontal temperature gradient or wind gradient? Which wind components in the horizontal? Be specific.
Pg. 23, Ln 12: "solves", not "is solving".

Pg. 23, Ln 14: "were", not "was".

Pg.23, Ln 15: "...their vertical gradients...", vertical gradient of which component?

Pg. 23, Ln 16-18: Which gradient is being referred to in these sentences, wind speed, temperature, and/or speed of sound?

Pg. 23, Ln 20-25: Why not talk about this when Figure 7 was introduced earlier?

Pg. 24, Ln 19: "controlled", does this mean measured?

Pg. 25, Ln 7-8: "Actually, no signal is receiving..." What is this sentence trying to say, it makes no sense.

Pg.25, Ln 9: What percent is "mostly possible" to measure an upwind-directed signal?

Pg. 25, Ln 10: "if the wind speed and therewith..." what is a "moderate gradient?" This sentence is awkward and needs to be clarified.

Pg. 25, Ln 23-24: "imply the main important inherent. . ." what influence is this referring to?

Pg. 35, Ln 23: "our results...maybe worth looking into." How else should the results be viewed and why weren't these ideas presented? It's possible a unique event occurred this night to produce a large advection value since the EC flux was large as well; this appears to be a case for analysis of more nights under various conditions to better vet the methodology.

Pg. 35, Ln 23: "Thereby, the different measurement volumes..." To what is this sentence referring? Feels out of place.

Pg. 35, Ln 24-26: "It is expectable that...". "Expected", not "expectable".

Pg. 35, Ln 26 and 28: Two Siebicke et al. papers are cited (2011 and 2012) but only one is present in the reference list.
Pg. 36, Ln 3: "to be on a safe side" should be "to be on the safe side".

Pg. 36, Ln 9: "will allow enhancing the security" could be "will enhance the security..."

Pg. 36, Ln 16: "has to take into account" should be "has to be taken into account".

Pg. 36; Ln 21-23: "Thus a highly...", This sentence would read better as "The results from a high number of optical and acoustic paths can be used..."

References: Feigenwinter, C., Bernhofer, C., Eichelmann, U., Heinesch, B., Hertel, M., Janous, D., Kolle, O., Lagergren, F., Lindroth, A., Minerbi, S., Moderow, U., Mölder, M., Montagnani, L., Queck, R., Rebmann, C., Vestin, P., Yernaux, M., Zeri, M., Ziegler, W., Aubinet, M., 2008. Comparison of horizontal and vertical advective CO2 fluxes at three forest sites. Agric. For. Meteorol. 148, 12–24. doi:10.1016/j.agrformet.2007.08.013

---

## Referee Comment (RC2) · Anonymous Referee #1 · 30 Jun 2017

**General comments:**

The scientific question of this manuscript is clear and this study is innovative. The paper is well structured in the sections of instrumentation, data analysis and uncertainty estimation. I would suggest an improvement of the Introduction section, and a further discussion about the temporal pattern of the advetion term on the basis of Fig. 18. Additionally, English language should be double checked.

**Specific comments:**

Introduction section

- Page 2 Line 8 says 'significant values', and Line 31 mentioned the threshold value of '5 $\mu$mol m-2 s-1'. It would be better to organized them into one paragraph.

- Page 2 Line 27 talks about the recent measurements of advection, while Page 3 Line 12 talks about the uncertainties of these measurements. It would be better to organized them into one paragraph.

- Page 3 Line 33. Separate the texts from 'Consequently' as a new paragraph.

English check, such as:

- Page 2 Line 17 should be 'with Vm as..., c as..., t as...'

- Page 2 Line 31. single sentence merge to a paragraph.

- Page 7 Line 21 to Page 8 Line 6. and Page 8 Line 30 should be written in past tense.

Page 8 Line 5: 'class-a-pan' should begin with capitalized letter.

- Page 8 Line 25: should be 'which was equipped'.

The two equations in Eq. (3) can be numbered separately.

FIg. 1 can be merged with Fig. 2.

---

## Author Comment (AC1) · 12 Aug 2017

The comment was uploaded in the form of a supplement:
https://www.atmos-meas-tech-discuss.net/amt-2017-94/amt-2017-94-AC1-supplement.zip

---

## Author Comment (AC2) · 12 Aug 2017

The comment was uploaded in the form of a supplement:
https://www.atmos-meas-tech-discuss.net/amt-2017-94/amt-2017-94-AC2-supplement.zip

---

## Author Response (AR1)

Referee #1

We would like to thank the anonymous reviewer #1 for valuable comments and support to improve the paper. In the revised manuscript, all these comments have been fully addressed, please see below. Comments of the reviewer are in black. Our responses are in red.

**General comments:**
The scientific question of this manuscript is clear and this study is innovative. The paper is well structured in the sections of instrumentation, data analysis and uncertainty estimation. I would suggest an improvement of the Introduction section, and a further discussion about the temporal pattern of the advetion term on the basis of Fig. 18. Additionally, English language should be double checked.
Revision/response: Thank you for the good assessment of our paper. The Introduction section was re-organized. Figs. with results of measurements were discussed more detailed. The English language was checked as well as possible. Furthermore, during the article processing of final revised papers for publication in AMT, English language copy-editing is included which will polish last shortcomings.

**Specific comments:**
Introduction section
- Page 2 Line 8 says 'significant values', and Line 31 mentioned the threshold value of '5 μmol m-2 s-1'. It would be better to organized them into one paragraph.
Revision/response: The sentences were organized into one paragraph.

- Page 2 Line 27 talks about the recent measurements of advection, while Page 3 Line 12 talks about the uncertainties of these measurements. It would be better to organized them into one paragraph.
Revision/response: The parts of text about measurements and uncertainties were organized into one paragraph.

- Page 3 Line 33. Separate the texts from 'Consequently' as a new paragraph.
Revision/response: The text was separated and organized as a new paragraph.

English check, such as:
- Page 2 Line 17 should be 'with Vm as..., c as..., t as...'
Revision/response: The sentence was corrected.

- Page 2 Line 31. single sentence merge to a paragraph.
Revision/response: Two sentences were merged and replaced to the first part of the Introduction.

- Page 7 Line 21 to Page 8 Line 6. and Page 8 Line 30 should be written in past tense.
Revision/response: We wrote the first sentence of this paragraph in Past Tense. We did not change the sentences 'Fluxes of CO2 (NEE), H2O (evapotranspiration) and sensible heat are available on a half-hourly basis.' and 'which are permanently updated according to sensor and software development'. The

EC measurements have been active since 2002 (permanent measurement station). Present Perfect is used here, because the action is continuing until now.

Page 8 Line 5: 'class-a-pan' should begin with capitalized letter.
Revision/response: It was corrected.

- Page 8 Line 25: should be 'which was equipped'.
Revision/response: It was corrected.

The two equations in Eq. (3) can be numbered separately.
Revision/response: The two equations were numbered separately.

FIg. 1 can be merged with Fig. 2.
Revision/response: The two Figs. were merged into Fig. 1.

Referee #2

We would like to thank the anonymous reviewer #2 for careful reading of the manuscript and thoughtful comments which allowed us to improve the revised paper. In the revised manuscript, all these comments have been fully addressed, please see below. Comments of the reviewer are in black. Our responses are in red. The changed text is additionally highlighted.

General Summary and Comments: Overall the paper is well structured and detailed, specially through Sections 2-4. Combining some of the figures into one group of figures would be useful (e.g. Figures 8 and 9 or Figures 14 to 18) since they are often talked about concurrently. It would also be beneficial to combine or summarize the uncertainties shown in Tables 2-4 with respect to the temperature and wind values for better readability. The results sections are light on details why the CO2 concentrations for the proposed approach are as different compared to the EC measurement (Specific Comments). Another sentence or two pointing toward the potential error sources and how it could be reduced would be useful for further studies. Overall well put together, but some minor revisions necessary.
There are spots where the writing needs to be cleaned up and clarification but these do not impede the reading of the work (See Technical Comments below).
Revision/response: Some of the figures were grouped together (in revised version: Figs. 13a, b, 14 a, b) to improve the comparability of results. The uncertainties used for further analysis were summarized in Table 4 and recalculated into wind and temperature uncertainties. Further specific and technical comments were treated below.

**Specific Comments:**
Please note: In the following, we explain the technical details of acoustic measurements. However, we did not add all the details into the manuscript because they are frequently out of scope. The explanations of the methods (A-TOM, OP-FTIR, soil chamber measurements), analysis, and results are included in the text of the revised manuscript.

Pg. 5, Ln 5-7: Are there not effects from wind passing from behind the microphones on the sound collection or are the microphones directional so this is not a concern?
Revision/response: We fully agree that the flow field around the microphone can influence the measurement of the sound signal, especially with respect to additional noise from the air flowing around the microphone. For this reason, we used wind screens for the microphones (please see text section 3.2.1 par. 1) to prevent flow-induced wind noise which disturb the received signal and reduce the signal-to-noise ratio (SNR). With the applied wind screens it is possible to receive a nearly undisturbed sound signal at the condenser microphone. Furthermore, the loudspeakers have a distinct directivity pattern. The locations of the microphones, which should receive a sound signal, are tailored to the directivity of the loudspeakers.

Pg. 9, Ln 17. What was the output power of the speakers?
Revision/response: The maximal output (sound pressure level) of the loudspeakers (in combination with an amplifier) is 100 dB.

Pg. 12, Ln 13-17: Can you provide more detail on the soil chamber measurement cycle? How long were the chambers closed for each interval and how was the concentration measured (see comment for Pg. 33, Figure 17).

Revision/response: The text was changed. Additional information explains the soil chamber measurements in more detail: The chambers installation was done one day before the data acquisition started to avoid any influences by disturbances due to the collar insertion. The obtained $CO_2$ data can be applied for the comparison with the spatially resolved GHG concentrations. The soil chamber measurements were done in accordance to ICOS protocol for automated chamber measurements (Pavelka and Acosta, 2016). We chose a sampling interval of two measurements per chamber per hour for the data acquisition period. An observation length of 120 s was chosen for the single soil flux measurements. Additionally, a pre-purge of 120 s and a post-purge of 45 s for each flux measurement were selected. The initial values of $CO_2$ concentration after the pre-purging and before the chamber closing were taken from the measured time series of the observation period for the determination of the considered $CO_2$ concentrations at the ground-level.

Pg. 13, Ln 5: "growing sound frequencies"; what does this mean with respect to the 7-kHz frequency used?

Revision/response: The used 7-kHz-signal meets the requirements of the signal analysis to produce the necessary accuracy in travel-time analysis. In general, the travel-time uncertainty decreases for increasing sound frequencies due to the process of signal analysis. It would be desirable to apply higher sound frequencies than 7 kHz. However, air absorption is a limiting factor which prevents the use of higher sound frequencies for the sound path distances under consideration (50-70 m). The sound absorption is about 8-9 dB/100 m for the used sound frequency of 7 kHz and typical values of meteorological quantities (DIN ISO 9613-1, 1993, temperature: 15 °C, relative humidity: 50 %, air pressure: 101325 Pa). It reaches values of more than 19 dB/100 m for a 10-kHz-signal which makes an application of high-frequency signals for acoustic sounding impossible at distances exceeding a few decameters (with the speakers we used.

The text was changed:

The used sound frequency is a compromise between the desired low travel-time uncertainty and the necessary high SNR. In general, the travel-time uncertainty is decreasing for increasing sound frequencies due to the process of signal analysis. Furthermore, higher frequencies allow for a high-pass filtering of received signals in order to exclude ambient low-frequency noise from data analysis which, in turn, enhances SNR. However, air absorption (see Sect. 4.1.1) is a limiting factor, which increases with increasing frequencies and thus prevents the use of arbitrarily high sound frequencies for the sound path distances under consideration. In view of additional acoustic ground effects (see Sect. 4.1.2), an optimal sound frequency of 7 kHz results for the investigated length scale up to 100 m. In view of additional acoustic ground effects (see Sect. 4.1.2), an optimal sound frequency of 7 kHz results for the investigated length scale up to 100 m.

Pg. 18, Ln 5-6: How much does the uncertainty decrease with the increased path lengths? If this is the case, why not use as long of a path length as possible instead of the "minimal path length of 50m"? What is the minimum path length that would generate usable data?

Revision/response: The uncertainty of temperature and wind depends on the path length $d$ according to the following equations (Eqs. 18 and 19 in the former manuscript):

$$\Delta T_{av} = 2\sqrt{\frac{T_{av}}{\gamma_{tr} R_{tr}}}\, \Delta\tau \left( \frac{(\gamma_{tr} R_{tr} T_{av}) + v_{Ray}^2}{d} \right)$$

$$\Delta v_{Ray} = \Delta\tau \left( \frac{(\gamma_{tr} R_{tr} T_{av}) + v_{Ray}^2}{d} \right)$$

Example: The uncertainty of temperature will decrease by the factor of 2 (about 0.3 K → 0.15 K, 0.2 m/s → 0.1 m/s) if the path length is doubled (50 m → 100 m).

The longest sound paths of the acoustic array (nearly a square) are about 71 m (diagonals of the square). To estimate the maximum uncertainty it is necessary to use the minimum path length, i.e. 50 m. That's why we used the wording 'minimal path length'.

Generally, it would be desirable to use longer paths to provide a minimum uncertainty of travel time. However, sound absorption and geometrical sound attenuation are increasing with growing path lengths followed by a decreasing SNR. Furthermore, the effects of sound reflection at the ground as well as refraction due to sound speed gradients have to be considered (see section 4.1.2). These phenomena of sound propagation prevent the application of longer sound paths with the used sound frequency of 7 kHz and the desired low travel-time uncertainty. If a lower sound frequency (with different loudspeakers, amplifiers, analog/digital converters…) is applied, then longer sound paths are possible (e.g., Ziemann et al., 2002).

The minimum path length depends on the application and the desired uncertainty of wind components and temperature. It depends also on the used hardware for analog/digital conversion and their possible resolution which depends on frequency.

We already used acoustic remote sensing for smaller field scales (e.g., Barth et al., 2013) or within a wind tunnel.

Reference:

Ziemann, A., K. Arnold, and A. Raabe, 2002: Acoustic Tomography as a Remote Sensing Method to Investigate the Near-Surface Atmospheric Boundary Layer in Comparison with In Situ Measurements. J. Atmos. Oceanic Technol., 19, 1208–1215, https://doi.org/10.1175/1520-0426(2002)019<1208:ATAARS>2.0.CO;2

Barth M, Fischer G, Raabe A, Weiße F, Ziemann A., 2013: Remote sensing of temperature and wind using acoustic travel-time measurements. Meteorologische Zeitschrift 22:103-109. doi: 10.1127/0941-2948/2013/0385

The text was changed:

For a travel-time accuracy of 78.125 μs and a path length of 50 m (minimum distance for the used geometry of sound paths), a maximum temperature uncertainty of about 0.3 K results for the instantaneous single path measurement.

Pg 18. Ln 22: Section 3.2.1 didn't mention any amplifiers. Were amplifiers used and how did you generate the sound wave for the speakers?

Revision/response: The sound wave is calculated by an own MATLAB script and generated by an acoustic multi-channel spectrometer card with four output channels (Harmonie PCI octav, SINUS Messtechnik GmbH, Germany, see section 3.3.1). Eight loudspeakers can be supplied with a signal by using this technique. The generated signal is delivered from the PC with the 'Harmonie' PCI card to an amplifier (Intersonic maxound mx 210). It amplifies the signal depending on the type of the connected loudspeaker (max. 100 dB for the used VISATON loudspeaker). Each of the Intersonic amplifiers has two separate channels. Two speakers are connected to each channel and are simultaneously supplied by the artificial signal. Overall, we used two amplifiers identical in construction.

Pg. 20, Ln 20-21: How stable was the frequency of the loudspeakers? How much of an error would this contribute to the results or was it negligible?

Revision/response: In our case, the influence of frequency stability of the loudspeaker is negligible. The high-end VISATON loudspeaker is characterized by a very high stability of frequency. The extremely light diaphragm made of a titanium-aluminum alloy has, in addition to an extremely dynamic pulse response, a very linear frequency response. If other loudspeakers will be applied, their frequency stability should be at least within a range of +/- 100 Hz around the desired signal frequency (see section 4.1.2 Reflection at ground surface).

Table 2: Could the corresponding wind and temperature errors be added to this table and Tables 3 and 4?

Revision/response: Table 3 contains real travel time differences between the arrival of the direct and the reflected sound rays. This travel time difference cannot be recalculated directly into temperature and wind uncertainties. A travel-time accuracy of 78.125 μs (= 4 samples/51.2 kHz) was applied for the further uncertainty analysis of sound speed, wind speed, and temperature for one instantaneous travel-time measurement along one sound path. This uncertainty of the measured travel time from the signal analysis (Table 2) and the uncertainties from the straight-ray approximation were summarized in Table 4. The corresponding wind and temperature errors were added in the revised version.
Table 4 was changed as follows:

**Table 1: Comparison of travel-time uncertainties: Above: Travel-time difference (in sample units), recalculated temperature and wind speed differences in brackets, between straight-line and curved sound path through the atmosphere for a maximum vertical gradient of effective sound speed of 0.6 $s^{-1}$ (during nighttime) on a summer day over grassland. Below: travel-time uncertainty (temperature and wind speed uncertainty in brackets) due to signal analysis using (CCF), see Sect. 4.1.1.**

| | Distance source-receiver in m | Height above ground in m |
|---|---|---|
| | | |

| | | 1.5 | 3.0 |
|---|---|---|---|
| Uncertainty due to travel-time difference between straight-line and curved sound path | 50.0 | 2
 (0.2 K; 0.1 ms$^{-1}$) | 0 |
| | 70.0 | 6
 (0.3 K; 0.2 ms$^{-1}$) | 1
 (0.1 K; 0.0 ms$^{-1}$) |
| Uncertainty due to signal analysis of travel time measurements | 50.0/70.0 | 4
 (0.3/0.2 K; 0.2/0.1 ms$^{-1}$) | |

Pg. 21, Ln 14-15: Overnight turbulence is not zero, there is still some minimal turbulence, but it may be negligible for this application. How much does turbulence negate the ground effects? Would this mean that daytime measurements are less error prone so better to validate the results of this method?

Revision/response: Salomons (2001) gives examples (especially numerical simulations) for the ground effect with and without turbulence. Thereby the turbulence was approximated by using von-Kármán spectrum of refractive-index fluctuations. Turbulence causes fluctuations of the instantaneous sound speed around the average value. Furthermore, turbulence causes fluctuations of the amplitude and phase of the sound waves travelling along the ray paths. The turbulent phase fluctuations are particularly important for the ground effect due to interference minima (interference of direct and reflected sound path). The interference minimum is considerably reduced by turbulence due to the phase fluctuations. The ground attenuation is limited to a value of about 20 -30 dB for maximum negative interference due to the influence of turbulence.

The SNR of the measured signal depends on the interference pattern due to the ground effect (depending on sound frequency, distances of loudspeaker and microphones, and heights of acoustic devices above ground surface). The influence of the 'ground dip' on the SNR is smaller if the atmosphere is more turbulent. That means that the requirements on frequency stability of loudspeakers and the geometry of the acoustic array are lower in comparison to atmospheric conditions with minimal turbulence. It is necessary to assume non-turbulent conditions to estimate the maximal uncertainty of the method.

We agree with the referee that daytime measurements are less error-prone considering this aspect of possible error sources. On the other side, the wind speed is normally increasing during daytime. This will lead to higher wind noise and a lower SNR. This effect is frequently higher than the ground effect, provided that an optimized sound frequency and acoustic array is applied for the measurements.

Pg. 27, Ln 6-8: Are the sonic temperatures the virtual sonic temperature or converted to ambient air temperature? The text refers to Tair and sonic temperature (_v).

Revision/response: We are aware of the difference between sonic temperature (= acoustic virtual temperature $=T_{av}$) and the ambient air temperature ($T_{air}$). Regarding the influences on sonic temperature we assumed relatively stable conditions concerning air humidity within 1-min intervals. Especially the variation of air temperature will have an impact on sonic temperature. That's why we considered the sonic temperature variability within 1-min intervals for the estimation of maximum air temperature error $\varepsilon_T$.

Pg. 28, Ln 3: Why was 20% chosen as the error level used?

Revision/response: Polak et al. (1995) stated important rules concerning temperature sensitivity and approximated requirements for passive measurements. As shown in Fig. 10 (revised manuscript) a reasonable absorbance errors lesser than 20 % can be achieved for an absolute value of $\varepsilon_T$ smaller than 0.4 K. The presentation of absorbance data in Fig. 11(b) (revised manuscript) shows the variability of absorbance and their increasing noise due to increased air temperature error $\varepsilon_T > 0.4$ K and decreased temperature differences $|T_B - T_{air}| < 2$ K. This consideration leads to the selection of time periods with absorbance errors lesser than 20 % (error range stated also in Polak et al., 1995).

Pg. 29, Ln 8: What constitutes "reasonable errors"?

Revision/response: The text was changed: Based on the previous data evaluation the absorbance spectra of the night time period from 10th – 11th July showed reasonable absorbance errors smaller than 20 % and were chosen for the subsequent quantitative analysis (Fig. 11b – revised manuscript).

Pg. 32, Figure 16: The FTIR CO2 concentrations are high compared to the EC measurements (Figure 17); is this a product of the line averaging or error within the FTIR/EC measurement systems?

Revision/response: There are only few experiences described in literature comparing point sensor and line-averaging ORS (Optical Remote Sensing) measurements. Furthermore, there are a few studies on ground-based ORS applications in micrometeorology targeting GHG fluxes. All references point out the limited comparability between OP-FTIR and point-scale measurements. These references report:
→ Differences in comparison of different line-averaging methods (e.g., OP-TDLAS, OP-FTIR) concerning retrieved path-integrated line concentrations (US EPA, 2011, von Bobrutzki et al., 2010, Thoma et al., 2005) for different gases
→ Experiments with a controlled release of target gases (e.g., CH4, C2H2 from a point source) representing mostly an underestimation of measured path-integrated concentrations (Polak et al., 1995, Reiche et al., 2014)
From our side, differences between the measured CO2 concentration (point-like and line-averaged) are expected due to the different volumes considered by the two different methods. OP-FTIR method used path lengths up to > 100 m including a field of view of about 10 mrad. As described in the answer to the comment 'Pg. 33, Figure 17', we assume heterogeneities in the spatial CO2 concentration due to heterogeneities in soil composition and soil respiration, slight topographic variability, variability in vegetation cover, etc. Furthermore, the measurements were carried out in different heights above ground (EC at 3 m, passive OP-FTIR Rapid at 0.9 m). Hence, the comparability of the point-scale

measurements of CO2 concentration at the EC tower with the OP-FTIR data is limited. A sophisticated quantification of differences between the point-scale and the line-averaging approach would require defined/artificial sources and/or a spatially distributed net of point-sensors to measure CO2 concentrations.


US-EPA, 2011. Optical remote sensing for measurement and monitoring of emissions flux. Handbook US environmental protection agency , Research Triangle Park.

Pg. 33, Figure 17: Soil chambers and EC CO2 measurements are not equivalent due to the height difference thus proximity to the source and the measurement style. At what point in the soil chamber cycle did the concentration measurement take place? Could you combine Figures 16 and 17 so the comparison between the concentration values were easier?

Revision/response: The variability in CO2 concentration measured by different techniques is not surprising. The main influences for the observed differences can be described by the different height of measurements above ground and the different measurement volumes.

Concerning the soil respiration data, the authors would like to give some comments using other references: Davidson et al. (2002) noted: Heterogeneity also exists within sites that appear mostly homogeneous to the investigator's eye. Hence the investigator is always faced with the question of how many chambers are needed to adequately estimate the mean and variance of CO2 fluxes within a site that is considered relatively homogeneous. Rodeghiero and Cescatti (2008) stated, that the variation of CO2 fluxes that is relevant to chamber measurements is often at the scale of centimeters, reflecting the sizes of rocks in soils, disturbances of soil fauna, pockets of fine root proliferation, and remnants of decaying organic matter. Whilst it is widely recognized that substrate availability, soil temperature and moisture largely influence the temporal and seasonal variability of soil CO2, the environmental factors controlling the spatial variability of CO2 effluxes are still poorly understood (Rodeghiero and Cescatti, 2008). Darenova et al. (2016) showed a spatial heterogeneity in soil flux for a grassland site of 17% (chamber measurements at a spatial scale of 50 m, 2x 1 week measurements, every 30 min for each chamber). Rochette et al. (1991) pointed out, that the determination of spatial pattern in soil respiration processes would help to interpolate between measurements and significantly reduce the number of sampling points required to estimate mean field values – a topic still under discussion up to now (Darenova et al., 2016).

The installation of the soil respiration chambers (SC) offered data from a distinct sampling point suitable to obtain an overview about the soil respiration as a source for atmospheric CO2 concentration.

Despite the spatial proximity of the two chambers to the EC tower, there are obviously differences in SC data itself as well as distinct differences in the temporal behaviour considering the comparison of EC data and SC data (former Fig. 17). Hence, the application of two soil respiration chambers placed nearby the EC station and with a distance of 5 m apart from each other at the edge of the observation field cannot be considered as a representative overview about the situation of the whole considered observation field (desirably homogeneous). However, the assumed heterogeneity in soil fluxes is influencing the optical measurements, although the OP-FTIR provides a spatial mean averaged across the optical line. Planned further investigations have to include more chamber measurements distributed in a wider spatial range to cover this aspect, but with respect for the potential disturbance due to additional equipment within the observation field.

See also comment to Pg. 12, Ln 13-17!

The (former) Figs. 16 and 17 were combined into one Figure with two parts (Fig. 14 a and b). On one hand it is possible to distinguish all graphs. On the other hand the concentration values can be easier compared within the new Fig. 15.

We totally agree, the generalized statement we formulated is not sustainable with the data shown here using only one point sensor (EC station) as verification and we have to change this passage. However, we need to express that with the available funds for the project an application / purchase of a distributed net of single point sensors was not possible. For further experiments this aspect will be included in experimental design, especially with the underlying question to observe potential spatial pattern in CO2 concentration and to quantify differences between the different methodical approaches of point-scale and line-averaging methods.

The text was changed as follows:

Obviously, a distinct similarity in concentration time series is observable for all measurements, but there are also significant differences concerning measured amplitudes of $CO_2$ concentration. The point measurements (SC and EC data) underlined the present variability in horizontal as well as in vertical distribution, also perceptible in OP-FTIR data. Furthermore, the chamber measurements at ground surface illustrated the increased spatial variability of $CO_2$ concentration during nighttime caused by soil respiration processes. Despite the spatial proximity of the two chambers to the EC tower, there are obviously differences in soil respiration data itself as well as distinct differences in the temporal behavior considering the comparison to the EC data. This spatial heterogeneity in soil flux for a grassland site can be caused by the variability in soil moisture, changes in soil fauna composition, and the amount of above-ground biomass (Davidson et al., 2002, Rodeghiero and Cescatti, 2008, Darenova et al., 2016). The data of Grillenburg experiment supports the approach of combined line-averaging and point measurements: OP-FTIR measurements provided path-integrated values covering assumed spatial concentration variability in a single measurement and yielded spatially averaged concentration values. However, a certainly limited comparability between results of point sensor and line-averaging measurements is expected due to the different volumes considered by the different methodical approaches and due to the effect of undersampling caused by the heavily limited number of point sensors.

Pg. 34, Ln 12: The range of uncertainty is very large and could change the sign of the horizontal advection values. Does the error range scale with the magnitude of the advection? How do we account for the uncertainty when comparing with EC-based advection?

Revision/response: The maximum uncertainty of the wind component is temporally constant. It depends only on the travel-time uncertainty and the distance. Compared with this, the maximum uncertainty of the horizontal gradient of CO2 concentration depends on the maximum uncertainties of the concentrations at the two optical paths (Fig. 16 of the former manuscript). These concentrations and therewith their uncertainties are temporally variable according to the derivation in section 4.2. The total maximum uncertainty of horizontal advection is calculated using the error propagation law (Eq. 25 of the former manuscript). The variable wind speed must also be included into this equation of maximum uncertainty. In this way, the error range scales with the magnitude of advection: smaller values with smaller values of advection and vice versa. The error estimation gives a maximum uncertainty of the advection derived by the proposed SQuAd method. The same strategy of error calculation should be worked out for the EC method (not only statistical uncertainties). Then it would be possible to compare the uncertainties of the two methods directly. It is a valuable advice of the referee and a perspective of a

future study, but out of the scope for the actual manuscript with focus on the applicability and uncertainty of SQuAd approach.

Pg. 34, Ln 18: What was the wind direction for this period? Did it remain constant or did it change with time? The change in sign of the v component in Figure 14 implies a changing wind direction and does this change the upstream source region for CO2 hence much larger advection?
Revision/response: The wind direction changed within the considered period according to Fig. 14 (in the former manuscript) from south-south-west directions until midnight up to westerly directions after 1 a.m. local time. The changing wind direction leads probably to another upstream source region for CO2, see Fig. 1. In southerly direction there is a greater area of grassland until reaching the forest. In westerly direction, a more heterogeneous surface is situated in about 200 m distance from the measurement array. This could also lead to changes in advection. Beside the upstream source region for CO2, the wind speed is the controlling factor of advection. The wind speed decreased noticeably together with the changed wind direction.
The estimation of the source area (also applying a boundary layer model) is a remaining task of the SQuAd project. The results of this study will be published in a future paper together with the data analysis of the whole measurement period.

Pg. 35, Ln 24-26: "It is expectable that. . .". My understanding is the differences in concentration measurements is a product of measurement height and measurement principle, not just line versus point measurements. Some different is expected from the line versus point measurement but there were points where the line average concentration (FTIR) doubled the point concentration (EC and SC) (Figures 16 and 17). This magnitude of difference in the concentration values doesn't seem to be just a difference in line versus point measurement averaging. I would expect the SC concentrations to be highest because of their proximity to the ground (CO2 source). Under-sampling of fluxes is different than under- or over-sampling the concentration values and the resulting advection values.
Revision/response: It is not surprising, that we discovered a correlation between initial values of near-ground atmospheric $CO_2$ concentration and the determined soil fluxes for the nighttime hours (see figure).

[Figure]

Hence, we assume similarities between soil flux heterogeneities and near surface $CO_2$ concentration pattern. Additionally, latter is also influenced by the slight topography and slightly variable vegetation parameters (e.g. grass species, slightly different height of vegetation). We agree that soil respiration is the main source for $CO_2$ in the atmosphere at the grassland site. However, due to the limited number of chambers the representativeness of the chosen location is not guaranteed for the whole field of observation. Many investigations in literature describe the interpolation uncertainties using chamber measurements by local effects (see also comment Pg. 33, Figure 17). Soil moisture variability is one of the most affecting parameters (e.g., Darenova et al., 2016). Actually, we observed differences in soil moisture data between the two chambers – in amplitude and also in temporal behaviour (see figure).

[Figure]

A detailed investigation of these site specific feedback processes is quite interesting and would require a long-term observation of soil CO2 efflux at different locations at Grillenburg site, which was not part of our project.

Changed text:

Thereby, the different measurement volumes of point-like and line-averaging measurement methods should be taken into account. We observed higher concentration values from spatially integrating and representative measurements in comparison to point measurements which could be affected by undersampling of real-world fluxes (Siebicke et al., 2011) and near-ground $CO_2$ concentration variability, too. The environmental factors driving the spatial variability of soil $CO_2$ fluxes are still poorly understood (Rodeghiero and Cescatti, 2008). Variability in physical soil properties (e.g., soil moisture, clay content), disturbances in soil fauna and the amount of above-ground biomass can produce spatial soil respiration heterogeneity also within a more or less homogeneous look alike grassland site (Davidson et al., 2002, Darenova et al. 2016). Hence, the spatial determination of GHG concentrations only based on point information requires an optimized vertically and horizontally distributed instrumental setup of point sensors. This is necessary for a representative site characterization avoiding the undersampling of the complex flow phenomena. Hence, the overarching application of line-averaging measurements can help to overcome the limitations of distributed single sensors providing integrative spatial data across an extended path less affected by local unrepresentative fluctuations.

Pg. 35, Ln 20: What is the error on the cited horizontal advection values; is the range you found similar to these studies?

Revision/response: Zeri et al. (2010) provided a standard error to the mean for their data (Fig. 2). The uncertainty (standard error to the mean) for horizontal advection is linked to the absolute (mean) value, but not with a simple linear dependency. Data sorted according to u* and averaged every 100 records. Considering only the nighttime data, the maximum value of horizontal advection amounts to 6.5 µmol m$^{-2}$ s$^{-1}$ with +/- 0.7 µmol m$^{-2}$ s$^{-1}$ for uncertainty.

The maximum uncertainty is about 1 µmol m$^{-2}$ s$^{-1}$.

Marcolla et al. (2014) used standard deviation of high-frequency measurements as a measure of uncertainty. The uncertainty of horizontal advection was dependent on the kind of measurement: a higher uncertainty (maximum about 7 µmol m$^{-2}$ s$^{-1}$) results without buffer volumes in comparison to measurements with added buffer volumes (maximum uncertainty about 2.5 µmol m$^{-2}$ s$^{-1}$).

We derived a maximum uncertainty which is higher than the data of these references due to the other method of calculation. The received values of uncertainties (3-38 µmol m$^{-2}$ s$^{-1}$), depending on the time and the amount of advection itself, are greater in comparison to an investigation of purely statistical uncertainties. Thereby, it is important to notice that we applied a maximum error calculation of the used methods A-TOM and passive OP-FTIR to be on a safe side for further applications, please see section 5 Conclusions and outlook.

**Technical Comments**

Pg. 1, Ln 22: "Thereby. . .", "Additionally" may be a better word.
Revision/response: The wording was changed according to the suggestion.

Pg. 1, Ln 27: Considering averaging. . ." does not need "Considering" and is missing a comma: "Averaging over a period of 30 minutes, the standard error".
Revision/response: The wording was changed according to the suggestion.

Pg. 2, Ln 2: "A closing gap for balance. . ." would be better as "Closing the gap for. . ."
Revision/response: We decided to let the text in the way it is because the wording satisfies the meaning.

Pg. 2, Ln 4: "measurement of flows" awkward phrasing
Revision/response: The wording was changed into 'flow measurements'.

Pg. 2, Ln 31: "considered advective fluxes" Vertical or horizontal advection?
Revision/response: The text was clarified:
Zeri et al. (2010) considered nighttime turbulent fluxes greater than 5 µmol m$^{-2}$ s$^{-1}$ as high values. … If vertical and/or horizontal advective $CO_2$ fluxes exceed such turbulent fluxes, then the advection influence can be considered as high.

Pg. 3, Ln 1: This sentence is misleading since substantial advection can occur in any land-cover type, the only requirement is a CO2 gradient to exist (e.g. Feigenwinter et al., 2008). I believe what the authors are attempting to convey is advection is commonly the largest error source in complex terrains.
Revision/response: The text was clarified:

Advection is a significant error source applying EC method mainly in complex terrain or in areas with land use changes (Aubinet, 2008).

Pg. 3, Ln 5-6: How much reduction in annual CO2 update? And in which forest?
Revision/response: One example is the Renon/Ritten, Italy, located at 1735 m. a.s.l. on a south exposed steep forested alpine slope (Feigenwinter et al., 2010) with a persistent slope wind system: The reported sink of 450 g C m $^{-2}$ for this forest (Norway Spruce with tree heights between 20 and 30 m, and a LAI of 5.5) is significantly reduced due to adding the (nighttime) advective fluxes to NEE. If total ecosystem respiration (TER) is estimated from the soil respiration, the advection corrected nighttime NEE would roughly increase by a factor of 1.8 and 3, dependent on the wind regime. These factors would be 2 until 6 for if TER is estimated from the sum of nighttime turbulent CO2 fluxes and changes in storage during well mixed conditions. The text was changed:
Taking such advective fluxes into account, a significant reduction of the reported annual $CO_2$ uptake of forests might be a feasible consequence (e.g. at the Renon/Ritten site, Feigenwinter et al., 2010).

Pg. 3, Ln 27: Remove extra parenthesis
Revision/response: It was changed according to the suggestion.

Pg. 5, Ln 19: Replace "several" with "two".
Revision/response: The wording was changed according to the suggestion.

Pg. 6, Ln 25: Don't need "Then,".
Revision/response: It was changed according to the suggestion.

Pg. 6, Ln 29: Not a complete sentence; rewrite.
Revision/response: It was changed ('are based on') to complete the sentence.

Pg. 7, Ln 18-19: Combine "The permanent EC. . ." and "Meanwhile is it. . ." into one sentence. Define "ICOS-D (C3 station)".
Revision/response: The text was changed:
The permanent EC station is working within FLUXNET since 2002 (e.g. Hussain et al., 2011a) and meanwhile within the network ICOS-D.

Pg. 8, Ln 10: "class-a-pan" should be "Class A pan".
Revision/response: It was changed.

Pg. 9, Ln 17: "speakers for frequencies" should be "speakers with frequencies".
Revision/response: It was changed according to the suggestion.

Pg. 14, Figure 6: Could you add the sent signal as well to illustrate the time shift?
Revision/response: The sent signal is shown in Fig. 5 (it starts at time 0 ms). In order to calculate the cross-correlation function (cp. Fig. 6, upper panel), the sent signal is shifted over the received signal

(Fig. 6, lower panel). The time shift, where the maximum of the cross-correlation function is achieved (= maximum accordance between the sent and the received signal) corresponds to the travel time of the signal.

Pg. 17, Ln 26: Need a comma between "50 m" and "a maximal".
Revision/response: It was changed.

Pg. 18. Ln 4: Remove "For", so "A minimal path length. . .".
Revision/response: It was changed according to the suggestion.

Pg. 19, Ln 16: "The latter is influenced. . .", don't need "one".
Revision/response: It was changed.

Pg. 21, Ln 5-10: This paragraph feels out of place. Some of this information was presented earlier in the section.
Revision/response: The text was deleted at this place and changed at section 4.1.2, par. 3:
In practice, the sound source and the receiver are close to the ground which makes sound propagation more complex. There are not only direct sound waves between loudspeaker and microphone, but also ground-reflected sound waves (Fig. 6). This wavelet integrates the conditions of the air layer between the ground surface and the receiver. Additionally, the interference between those sound waves can lead to considerable effects which are estimated hereafter.

Pg. 21, Ln 16-17: "It was proven, whether. . .", which of the two were proven, that you can separate the two sound wave parts or it wasn't possible to separate the two parts?
Revision/response: The text was clarified:
It was examined whether the directly propagating and the reflected sound wave parts could be separated due to their time delay at the receiver.

Pg. 22, Ln 9-10: Repeat from earlier.
Revision/response: The text was changed:
Furthermore, the real measurement height of the acoustically derived wind velocity and temperature can be slightly smaller than the geometrical height of the acoustic devices above ground because the received signal contains partially the properties of the atmospheric layer between ground surface and microphone.

Pg. 22, Ln 22-23: ". . .vertical wind and temperature gradients. . ."; vertical gradient of the horizontal wind? And what "horizontal ones"? Horizontal temperature gradient or wind gradient? Which wind components in the horizontal? Be specific.
Revision/response: The wording was changed to specify wind components and gradients: 'Thereby, vertical gradients of horizontal wind velocity and temperature are especially important because they are usually greater than associated horizontal gradients.'

Pg. 23, Ln 12: "solves", not "is solving".
Revision/response: It was changed.

Pg. 23, Ln 14: "were", not "was".
Revision/response: It was changed.

Pg.23, Ln 15: ". . .their vertical gradients. . .", vertical gradient of which component?
Revision/response: The wording was changed to specify vertical gradients: 'Calculation of temperature, wind velocity, and humidity profiles were followed by a calculation of the effective sound speed and its vertical gradients as average over 30 min for several local times (Fig. 9). At the transmitter height of 1.5 m or 3 m, positive vertical gradients of effective sound speed can be expected for a sound propagation in wind direction.'

Pg. 23, Ln 16-18: Which gradient is being referred to in these sentences, wind speed, temperature, and/or speed of sound?
Revision/response: It is the gradient of effective sound speed: 'At the transmitter height of 1.5 m or 3 m, positive vertical gradients of effective sound speed can be expected for a sound propagation in wind direction.'

Pg. 23, Ln 20-25: Why not talk about this when Figure 7 was introduced earlier?
Revision/response: The main content of (former) Fig. 7 is to illustrate the effect of sound reflection at the ground surface (paths of sound waves propagating above a ground surface). This effect is described in the paragraph that follows the Figure. The effect of sound refraction is described later in this section. For this reason the (former) Fig. 7 is placed at the beginning of the section describing the ground reflection. A back reference is given later to this figure.

Pg. 24, Ln 19: "controlled", does this mean measured?
Revision/response: The wording was changed into 'measured'.

Pg. 25, Ln 7-8: "Actually, no signal is receiving. . ." What is this sentence trying to say, it makes no sense.
Revision/response: The wording was changed to explain that theoretically no sound ray would reach the microphone for upwind conditions: For such conditions, theoretically no signal reaches the microphone which is located at the same height level as the loudspeaker but several decameters away from the speaker.
Nevertheless, due to a finite extent of the microphone, its spherical directional pattern and due to the scattering effect of atmospheric turbulence (Salomons, 2001), it is almost always possible to detect a signal in upwind direction if the wind speed is smaller than 6 m s$^{-1}$ at a height of acoustic devices and therewith the vertical gradient is moderate (around 0.3 s$^{-1}$).

Pg.25, Ln 9: What percent is "mostly possible" to measure an upwind-directed signal?
Revision/response: The wording was changed, see comment Pg. 25, Ln 7-8.

Pg. 25, Ln 10: "if the wind speed and therewith. . ." what is a "moderate gradient?" This sentence is awkward and needs to be clarified.
Revision/response: The text was clarified, see comment Pg. 25, Ln 7-8.

Pg. 25, Ln 23-24: "imply the main important inherent. . ." what influence is this referring to?
Revision/response: Infrared spectral data are mainly controlled by the environmental conditions such as pressure and temperature variations.

Pg. 35, Ln 23: "our results. . .maybe worth looking into." How else should the results be viewed and why weren't these ideas presented? It's possible a unique event occurred this night to produce a large advection value since the EC flux was large as well; this appears to be a case for analysis of more nights under various conditions to better vet the methodology.
Revision/response: The main focus of the paper lies on the description of the line-averaging methods and the detailed derivation of uncertainties of the SQuAd approach. First results of measurements were also presented, discussed and compared with values from literature. We agree with the referee that presentation and discussion of advection results is not complete (e.g. comparison of different nighttime measurement, complete data analysis), but this was out of the papers scope. The interest of the referee strongly motivates us to prepare a further paper presenting all results of the measurement campaign. The text was changed:
In this respect our results at relatively flat grassland site and using the line-averaging methods are worthy of discussions. Thereby, the different measurement volumes of point-like (measurements based on EC) and line-averaging measurement methods (OP-FTIR, A-TOM) should be taken into account. We observed higher concentration values from spatially integrating and representative measurements in comparison to point measurements which could be affected by undersampling of real-world fluxes (Siebicke et al., 2011) and near-ground $CO_2$ concentration variability, too. The environmental factors driving the spatial variability of soil $CO_2$ fluxes are still poorly understood (Rodeghiero and Cescatti, 2008). Variability in physical soil properties (e.g., soil moisture, clay content), disturbances in soil fauna and the amount of above-ground biomass can produce spatial soil respiration heterogeneity also within a more or less homogeneous look alike grassland site (Davidson et al., 2002, Darenova et al. 2016). Hence, the spatial determination of GHG concentrations only based on point information requires an optimized vertically and horizontally distributed instrumental setup of point sensors. This is necessary for a representative site characterization avoiding the undersampling of the complex flow phenomena. Hence, the overarching application of line-averaging measurements can help to overcome the limitations of distributed single sensors providing integrative spatial data across an extended path less affected by local unrepresentative fluctuations.

Pg. 35, Ln 23: "Thereby, the different measurement volumes. . ." To what is this sentence referring? Feels out of place.
Revision/response: The text was changed, see comment Pg. 35, Ln 23.

Pg. 35, Ln 24-26: "It is expectable that. . .". "Expected", not "expectable".
Revision/response: The wording was changed.

Pg. 35, Ln 26 and 28: Two Siebicke et al. papers are cited (2011 and 2012) but only one is present in the reference list.
Revision/response: The reference Siebicke et al. (2011) was added in the reference list.

Pg. 36, Ln 3: "to be on a safe side" should be "to be on the safe side".
Revision/response: The wording was changed.

Pg. 36, Ln 9: "will allow enhancing the security" could be "will enhance the security. . ."
Revision/response: The wording was changed.

Pg. 36, Ln 16: "has to take into account" should be "has to be taken into account".
Revision/response: The wording was changed.

[revised manuscript text omitted]

---

## Author Response (AR2)

Associate Editor

In the revised manuscript, all comments have been fully addressed, please see below. Comments of the Associate Editor are in black. Our responses are in red.

1) reference list
- please add DOIs to all the entries in your reference list. I understand that some of the references might not have a DOI but all the ones from scientific journals should have one. If there is no DOI but the reference is available online, please add a URL.
Where possible, we added DOIs. Otherwise we added ISBN number or URL.

- while you are at it: the abbreviated journal names in your reference list do not seem to follow standard rules. E.g. "J Quant Spectrosc Ra" instead of "J. Quant. Spectrosc. Radiat. Transf.". The copy editor will likely complain about this too.
We asked the team of Copernicus Publications (Sarah Schneemann, Typesetting | Team Coordinator). They sent us the following information: "Yes, the link you used (http://library.caltech.edu/reference/abbreviations/) is the actual one which will also be used during the typesetting process, so please follow these abbreviations."
We double-checked all used journal abbreviations according to the ISI Journal Title Abbreviations (http://library.caltech.edu/reference/abbreviations/). As stated there, the abbreviation for the "Journal of quantitative spectroscopy & radiative transfer" should be "J Quant Spectrosc Ra". But there are two other journal abbreviations which have been corrected: Agr Forest Meteorol and Appl Acoust

2) missing sections on data availability and author contributions
According to the AMT "Manuscript preparation guidelines for authors"
https://www.atmospheric-measurement-techniques.net/for_authors/manuscript_preparation.html
all manuscripts should include sections on data availability (required) and author contributions (recommended). I ask you to have another look at the manuscript preparation guidelines and add these missing sections.
- Data sets: Authors are requested to follow our data policy including the deposit of data that correspond to journal articles in reliable data repositories, the assignment of digital object identifiers, and the proper citation of a data set. Authors are required to provide a statement on how their underlying research data can be accessed. This must be placed as the section "Data availability" at the end of the manuscript before the acknowledgements. If the data are not publicly accessible, a detailed explanation of why this is the case is required.
Note: It is fine if your data is only available on request. However, this should be stated

explicitly.

We added a section on data availability according to the AMT guidelines.

- Author contribution: Authors are encouraged to add a section "Author contribution" before the acknowledgements in which the contributions of all co-authors are briefly described. Example: AA and BB designed the experiments and CC carried them out. DD developed the model code and performed the simulations. AA prepared the manuscript with contributions from all co-authors.

We added a section on author contribution according to the AMT guidelines.

[revised manuscript text omitted]

10 **Data availability**

Data are available upon request by the corresponding author at present. The data sets will be freely available on servers after finishing all analysis within the SQuAd project. Please follow the updates on the project web sites for access information: https://tu-dresden.de/bu/umwelt/hydro/ihm/meteorologie/forschung/forschungsprojekte/spatial/index.

**Appendix A**

15 Using the assumption of reciprocal sound propagation (see Eqs. (11) and (12)), it follows for the uncertainty of the acoustic virtual temperature $\Delta T_{av}$ and wind component along sound path $\Delta v_{Ray}$:

$$\Delta T_{av} = \left(\left|\frac{\partial T_{av}}{\partial d}\right| \Delta d\right) + \left(\left|\frac{\partial T_{av}}{\partial \tau_{forth}}\right| \Delta \tau_{forth}\right) + \left(\left|\frac{\partial T_{av}}{\partial \tau_{back}}\right| \Delta \tau_{back}\right) \text{ and} \tag{A1}$$

$$\Delta v_{Ray} = \left(\left|\frac{\partial v_{Ray}}{\partial d}\right| \Delta d\right) + \left(\left|\frac{\partial v_{Ray}}{\partial \tau_{forth}}\right| \Delta \tau_{forth}\right) + \left(\left|\frac{\partial v_{Ray}}{\partial \tau_{back}}\right| \Delta \tau_{back}\right). \tag{A2}$$

Differential measurements outgoing from a known initial state increase the accuracy because errors of the path length

20 measurement can be compensated. With $\Delta d = 0$ it follows from Eqs. (A1) and (11):

$$\Delta T_{av} = \left(\left|\frac{\partial T_{av}}{\partial \tau_{forth}}\right| \Delta \tau_{forth}\right) + \left(\left|\frac{\partial T_{av}}{\partial \tau_{back}}\right| \Delta \tau_{back}\right) = \frac{1}{\gamma_d R_d} \frac{d^2}{2} \left(\frac{1}{\tau_{forth}} + \frac{1}{\tau_{back}}\right) \left(\frac{1}{\tau_{forth}^2} \Delta \tau_{forth} + \frac{1}{\tau_{back}^2} \Delta \tau_{back}\right) \tag{A3}$$

Assuming that travel-time errors along one and the same path in opposite directions (forth and back) are identical to $\Delta \tau$, the temperature uncertainty from Eq. (A3) can be written:

$$\Delta T_{av} = \frac{1}{\gamma_d R_d} \frac{d^2}{2} \Delta \tau \left(\frac{2\sqrt{\gamma_d R_d T_{av}}}{d}\right) \left(\frac{2(\gamma_d R_d T_{av}) + 2 v_{Ray}}{d^2}\right). \tag{A4}$$

25 The uncertainty of relative wind measurements is only depending on the uncertainty of travel-time measurements:

$$\Delta v_{Ray} = \left(\left|\frac{\partial v_{Ray}}{\partial \tau_{forth}}\right| \Delta \tau_{forth}\right) + \left(\left|\frac{\partial v_{Ray}}{\partial \tau_{back}}\right| \Delta \tau_{back}\right) = \frac{d}{2} \left(\frac{1}{\tau_{forth}^2} \Delta \tau_{forth} + \frac{1}{\tau_{back}^2} \Delta \tau_{back}\right). \tag{A5}$$

**Author contribution**

A. Ziemann (AZ) and M. Starke (MS) are responsible for A-TOM and C. Schütze (CS) for OP-FTIR. All authors designed the SQuAd campaign and carried out the experiment. The overall co-ordination was carried out by AZ. MS developed and performed the code for controlling A-TOM and analysing acoustical data. CS developed and performed the code for controlling OP-FTIR and analysing optical data. AZ prepared the joint data analysis of A-TOM, sonic and EC data together with the uncertainty calculation of line-averaged wind components (A-TOM). CS prepared the line-averaged concentration data (OP-FTIR) with an uncertainty analysis. AZ prepared the manuscript with contributions from all co-authors.

**Competing interests**

The authors declare that they have no conflict of interest.

[revised manuscript text omitted]